# Snow depth product over Antarctic sea ice from 2002 to 2020 using multisource passive microwave radiometers

Xiaoyi Shen[1,2], Chang-Qing Ke[1,2], Haili Li[1,2]

[1] School of Geography and Ocean Science, Nanjing University, Nanjing, 210023, China

[2] Jiangsu Provincial Key Laboratory of Geographic Information Science and Technology, Nanjing University, Nanjing, 210023, China

*Correspondence to*: Chang-Qing Ke (kecq@nju.edu.cn)

**Abstract.** Snow over sea ice controls energy budgets and affects sea ice growth/melting, and thus has essential effects on the climate. Passive microwave radiometers can be used for basin-scale snow depth estimation at a daily scale; however,

previously published methods applied to the Antarctic clearly underestimated snow depth, limiting their further application. Here, we estimated snow depth using passive microwave radiometers and a newly constructed, robust method by incorporating lower frequencies, which have been available from AMSR-E and AMSR-2 since 2002. A regression analysis using 7 years of Operation IceBridge (OIB) airborne snow depth measurements showed that the gradient ratio (GR) calculated using brightness temperatures in vertically polarized 37 and 7 GHz, i.e., GR(37/7), was optimal for deriving

Antarctic snow depth, with a correlation coefficient of -0.64. We hence derived new coefficients based on GR(37/7) to improve the current snow depth estimation from passive microwave radiometers. Comparing the new retrieval with in situ measurements from the Australian Antarctic Data Centre showed that this method outperformed the previously available method (i.e., linear regression model based on GR(37/19)), with a mean difference of 5.64 cm and an RMSD of 13.79 cm, compared to values of -14.47 cm and 19.49 cm, respectively. A comparison to shipborne observations from Antarctic Sea Ice

Processes and Climate indicated that in thin ice regions, the proposed method performed slightly better than the previous method (with RMSDs of 16.85 cm and 17.61 cm, respectively). We generated a complete snow depth product over Antarctic sea ice from 2002 to 2020 on a daily scale, and negative trends could be found in all sea sectors and seasons. This dataset (including both snow depth and snow depth uncertainty) can be downloaded from National Tibetan Plateau Data Center, Institute of Tibetan Plateau Research, Chinese Academy of Sciences at http://data.tpdc.ac.cn/en/disallow/61ea8177-7177-

4507-aeeb-0c7b653d6fc3/ (Shen and Ke, 2021, DOI: 10.11888/Snow.tpdc.271653).

## 1 Introduction

Snow is a basic element in the Antarctic sea ice system and it changes the surface albedo of sea ice (Petrich et al., 2012), controls energy exchanges between the atmosphere and ocean (Kwok and Untersteiner, 2011) and affects sea ice growth and melting (Maykut et al., 1971; Sturm et al., 2002). Thus, it has essential climatic effects (Webster et al., 2018). Because snow

depth is a fundamental property of snow cover, knowing how it changes is crucially important for understanding rapid

changes in the Antarctic climate. Snow depth is also an essential input for sea ice thickness estimation (Giles et al., 2007; Kwok et al, 2020), and its accuracy will greatly affect the reliability of sea ice thickness estimates. Hence, from the perspectives of climate and sea ice thickness estimation, basin-scale snow depth products in the Antarctic, especially over a long time period, are urgently needed.

Although in situ measurements of snow depth over Antarctic sea ice have very high accuracy and precision, their spatial and temporal coverage are quite limited. Airborne snow depth measurements can cover regions of thousands of square kilometres, but they are cost intensive and represent only limited regions. Only satellites can obtain snow depth at the hemispheric scale, and individual and multisource satellites have been applied for snow depth estimation, e.g., passive microwave radiometers (Markus and Cavalieri, 1998; Comiso et al., 2003; Maaß et al., 2013), satellite radar altimeters
(Guerreiro et al., 2016; Lawrence et al., 2018), satellite laser altimeters (Kern et al., 2016), and a combination of satellite radar and laser altimeters (Kwok et al., 2019; Kacimi and Kwok, 2020). Given both the basin-scale coverage and the temporal resolution requirements, passive microwave radiometers are the best tools to derive a long data record of snow depth in the Antarctic with daily coverage.

The theoretical basis of snow depth estimation from passive microwave radiometers is that the volume scattering of upper
snow cover affects the radiation signal emitted from the underlying sea ice and reduces the observed brightness temperatures (Markus and Cavalieri, 1998). Thus, the observed brightness temperatures are related to the observation frequency and snow depth, and the snow brightness temperature increases as snow depth decreases or observation frequency increases. Based on this principle, Markus and Cavalieri (1998) used correlation analysis for the measured snow depth and brightness temperatures observed from the Special Sensor Microwave/Imager (SSM/I) in the Antarctic. They found that the gradient
ratio (GR) calculated from vertical polarized brightness temperatures at 19 GHz and 37 GHz had the highest correlation with measured snow thickness with a correlation coefficient of -0.60. An empirical linear regression equation was then derived for snow depth estimation, and the regression coefficients were updated for the successor passive microwave ratiometer (i.e., Advanced Microwave Scanning Radiometer for EOS (AMSR-E), Comiso et al., 2003).

Although this method can derive basin-scale snow depth, due to the snow penetration depth when 37 and 19 GHz
frequencies (i.e., higher frequencies) are used and the strong influence liquid water in the snow layer has on the observed radiation from passive microwave ratiometers, this method is limited to dry snow less than 50 cm thick and thus may underestimate the snow depth in some regions of the Antarctic. Given these influences, this method obviously underestimates snow depth by a factor of 2.3 (Worby et al., 2008a) or between 2 and 4 (Kern et al., 2011). Since 2002, successful launches of AMSR-E and its successor Advanced Microwave Scanning Radiometer 2 (AMSR-2) have provided a
chance to estimate snow depth with lower frequencies. Lower frequencies are sensitive to deeper ice layers, are less affected by liquid water in the snow layer and weather conditions (Rostosky et al., 2018) and have been used to improve snow depth estimation over Arctic sea ice (Rostosky et al., 2018; Braakmann-Folgmann et al., 2019; Kilic et al., 2019; Winstrup et al., 2019). Compared to the Arctic, snow depth over Antarctic sea ice is usually thicker (Kern et al., 2016), more heterogeneous

(Massom et al., 2001) and less affected by surface melting; hence, lower frequencies tend to be more suitable for retrieving Antarctic snow depth. However, these methods have not been tested or applied to Antarctic snow depth estimation until now.

In the present study, we attempt to construct a new and effective method to estimate snow depth over Antarctic sea ice. For the potential improvement of snow depth estimation using low-frequency signals, AMSR-E and AMSR-2 were used to derive new regression coefficients in the estimation equation. A detailed introduction to these data is shown in Section 2. Section 3 describes the methods for snow depth and uncertainty estimations, and the accuracy evaluation is shown in Section 4. Section 5 shows the spatiotemporal variation in the derived Antarctic snow depth between 2002 and 2020. Section 6 discusses the uncertainty sources of the proposed method, Section 7 gives the data availability and Section 8 concludes this paper.

## 2 Data

### 2.1 AMSR-E, AMSR-2 and SSMIS brightness temperature observations

To generate a complete time series of snow depth data over Antarctic sea ice, multiple passive microwave radiometers were used, including AMSR-E, AMSR-2 and the Special Sensor Microwave Imager Sounder (SSMIS). Between 1 June 2002 and 30 September 2011, AMSR-E data were used. With an observation angle of 55°, AMSR-E can provide daily brightness temperature observations in the whole Arctic and Antarctic. Six frequency channels were applied, i.e., 6.93, 10.7, 18.7, 23.8, 36.5 and 89.0 GHz, and each channel had both horizontal and vertical polarizations. Here, the AMSR-E/Aqua Daily L3 25 km Brightness Temperature Polar Grids (Version 3) product from the National Snow and Ice Center (NSIDC) were used, and pre-processing, bias correction and quality control were all applied (Cavalieri et al., 2014).

Between 2 July 2012 and 31 May 2020, AMSR-2 data were used. Compared to AMSR-E, AMSR-2 has the same observation angle and frequency channels but has an additional frequency at 7.3 GHz. Here, the NSIDC AMSR-E/AMSR-2 Unified L3 Daily 25 km Brightness Temperature Polar Grids (Version 1) product was used, and pre-processing, bias correction and quality control were also applied (Markus et al., 2018).

Brightness temperature observations from SSMIS were used to fill the gap in AMSR-E and AMSR-2 data between 1 October 2011 and 1 July 2012. The DMSP SSM/I-SSMIS Daily Polar Gridded Brightness Temperature (Version 4) product was used here because it has the same spatial/temporal resolutions and spatial coverage as the two brightness temperature products mentioned above, pre-processing, bias correction and quality control were also applied (Maslanik et al., 2004). All three passive microwave radiometers can provide the daily brightness temperature observations for the whole Antarctic. The temporal coverage of this dataset is from 14 December 2006 to 31 March 2019, hence it can be used to fill the observation gap between AMSR-E and AMSR-2 data and has a long overlapped time period with AMSR-E/2. However, SSMIS does not have lower frequency channels (than 19 GHz); hence, the corresponding snow depth estimation equation was adjusted accordingly, as shown in Section 3.

To generate consistent brightness temperature observations from 2002 to 2020, a consistency correction should be applied to the three passive microwave ratiometer datasets. The SSMIS brightness temperature observations were calibrated to AMSR-E data based on the method from Wentz (2013), and we calibrated the AMSR-2 data to AMSR-E based on the correction parameters from Du et al. (2014). These brightness temperature observations from AMSR-E, AMSR-2 and SSMIS were also used to obtain the full time (2002-2020) sea ice concentrations by using the ARTIST Sea Ice (ASI)

algorithm (Spreen et al., 2008).

## 2.2 Operational IceBridge airborne snow depth measurements

The initial aim of the Operational IceBridge (OIB) airborne mission is to fill the observation gap between ICESat and ICESat-2. This mission provides annual measurements of snow depth over sea ice, elevation and thickness of sea ice, and information on sea ice types in the Arctic and Antarctic. Due to the large coverage of measurements, it was suitable to

evaluate satellite-derived parameters. In the OIB airborne mission, Airborne Topographic Mapper (ATM), a laser altimeter, is used to measure the elevation of the sea ice surface. Its footprint depends on the observation angle of the pulsed laser and flight altitude. The size is approximately 1 m (Kurtz et al., 2013), and the location and elevation measurements accuracies for individual measurements are approximately 1 m and 0.1 m (or better, Krabill et al., 1995; Schenk et al., 1999), with a vertical precision of 3 cm (Martin et al., 2012).

The measured elevations were used to derive the total freeboard. Following Zwally et al. (2008) and Kern et al. (2015), the lowest 2% elevations in a 50 km segment along the track were regarded as the sea surface heights, and the mean value was calculated as the mean sea surface segment height (MSSH). Other points were taken as sea ice measurement points, and the corresponding total freeboard was calculated by subtracting the local MSSH from the sea ice surface heights.

    Snow radar is used to measure the snow depth in the OIB airborne mission. However, for snow cover over Antarctic sea

ice, the snow-ice interface is hard to distinguish (Giles et al., 2008; Willatt et al., 2009) due to the complicated snow morphology often found in the Antarctic (Massom et al., 2001). Accurate snow depth detection needs more in situ investigations and in-depth studies. Considering these influences, snow depth was derived from the total freeboard as described in Ozsoy-Cicek et al. (2013). The corresponding linear equations were constructed in six individual sea sectors in the Southern Ocean, with correlation coefficients ranging from 0.81 to 0.99. These have been widely used in previously

published studies to obtain Antarctic sea ice parameters (Xie et al., 2013; Kern et al., 2016; Li et al., 2018).

    OIB ATM data collected in 2009-2014, and 2016–2018 were used, no data could be obtained for the year of 2015. OIB data in 2011 were not used for the derivation of snow depth estimation equation to reduce the potential effect of the inter-mission calibration between SSMIS and AMSR-E/2, but they were used for the independent evaluation of SSMIS-derived snow depth in 2011. In each year except 2013 the OIB ATM data were acquired in both October and November, for the 2013

OIB campaign only measurement data in November were obtained, more details about the time information of OIB data are available via https://nsidc.org/data/ILATM2/versions/2. The spatial distributions of the used OIB ATM data (after data filter, see Section 3.1) are shown in Fig. 1a. Most of the OIB ATM data came from the western of Antarctic sea ice region and one

track covered the Ross Sea sector. These measurements covered both the thicker snow in the Weddell Sea sector and the thinner snow in the Ross Sea sector and provided comprehensive measurements for the development of satellite-based snow depth estimation methods. For comparison purposes, the OIB snow depth measurements were averaged in the overlapped passive microwave radiometer grid cells (at the spatial resolution of 25 km) in the same day. Although the sea ice is continuously drifting, the time differences between the OIB and passive microwave ratiometer data were always less than one day, which can cause the sea ice drift of several kilometers. Comparing to the coarse spatial resolution of passive microwave ratiometers (i.e., 25 km), this effect can be ignored. More details can be referred to Section 3.1, this processing method was also applied for other data sets as listed in Sections 2.3 and 2.4.

## 2.3 AADC in situ measurement data

We used in situ snow depth measurements from the Australian Antarctic Data Centre (AADC) to evaluate the proposed method. AADC in situ data include measurements of sea ice and snow from 1985 to 2007. This dataset provides records of snow depth, sea ice freeboard and sea ice thickness. Here, AADC data between September and October 2003, between September and October 2007 were used to compare our snow depth estimation results, which were mainly located in eastern and western of the Antarctic sea ice region (Fig. 1b, only the used AADC data (after data filter, see Section 4.2) are shown), more details about the time information of AADC data can be available via https://data.aad.gov.au/metadata/records/sea_ice_measurements_database. Although in situ measurements are relatively rare, AADC has measurements of both thick and thin ice, which provide a comprehensive and accurate evaluation of estimated snow depth.

## 2.4 ASPeCt shipborne observation data

We also used snow depth observations from the Antarctic Sea Ice Processes and Climate (ASPeCt) mission to evaluate the estimated snow depth. These data (including observations of snow depth, sea ice thickness and ice type) were obtained every hour within a 1 km radius of the ship. We followed the Worby et al. (2008b) method to reduce the sampling bias caused by temporal data collection and variable ship speed by removing observations within 6 nautical miles of previous observations. This method ensured the independence of each record. As the passive microwave ratiometer observes both undeformed and deformed sea ice, the 'averaged snow depth' record was used to compare to the passive microwave ratiometer-derived snow depth. According to the error analysis in Worby et al. (2008b), ± 20% bias of ASPeCt data is found for undeformed ice thicker than 0.3 m, and ± 30% bias is found for deformed ice.

This dataset contains snow depth measurements from 81 cruises between 1981 and 2005. Here, we used ASPeCt data between 2002 and 2005, which covered various types of sea ice and most sea sectors in the Southern Ocean (Fig. 1b, only the used ASPeCt data (after the data filter, see Section 4.3) are shown), more details about the time information of ASPeCt data are available via http://aspect.antarctica.gov.au/data.

## 2.5 ICESat-2 data

ICESat-2 data were used here to estimate the snow depth over Antarctic sea ice and the estimated snow depth was compared to estimates from the proposed method. Kern et al. (2016) found that satellite laser altimeters can be used to estimate snow depth over Antarctic sea ice with a low level of uncertainty, and these snow depth measurements agreed closely with both shipborne and airborne data. Considering the potential reliability of satellite laser altimeter-derived snow depth, following Kern et al. (2016), we estimated the Antarctic snow depth in a complete year (January 2019 to December 2019) from

ICESat-2 using a linear equation based on total freeboard (Ozsoy-Cicek et al., 2013). The ICESat-2 ATL10 sea ice product (Kwok et al., 2019a), which contains total freeboard estimates, was used for snow depth estimation. The along-track resolution of the total freeboard estimates is variable and is determined by the number of pulse footprints to aggregate the 150 photons. For strong beams, this along-track resolution varies between ~10 m and 200 m, and it varies between ~40 m and 800 m for weak beams. An averaged bias of 2–4 cm for ICESat-2 ATL10 total freeboard was found based on assessment

in Kwok et al. (2019b). The detailed algorithm for the ICESat-2 total freeboard estimates can be found in Kwok et al. (2019c). ICESat-2 ATL10 sea ice product for the Antarctic sea ice between January and December 2019 were used in the present study.

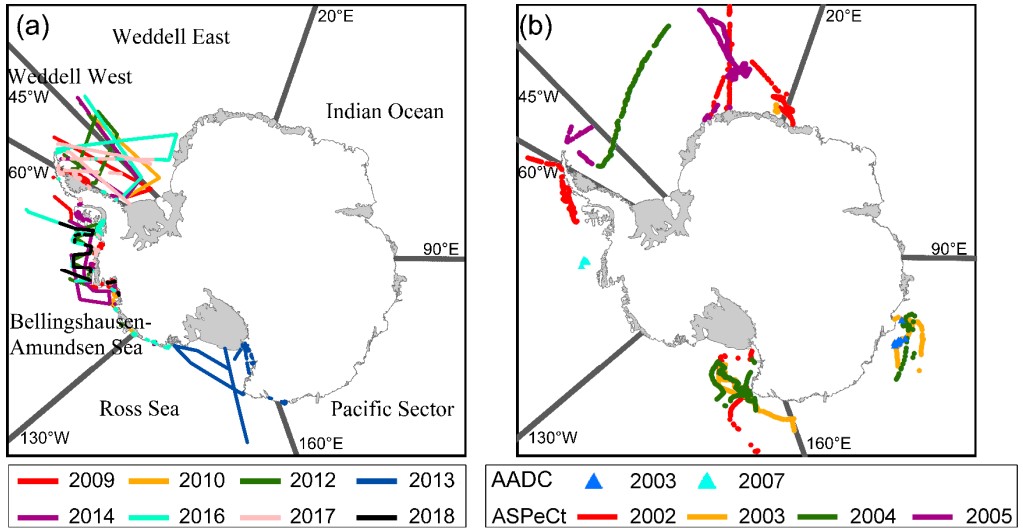

**Figure 1**. The spatial-temporal distributions of the used OIB airborne measurements (a), AADC in situ measurements and ASPeCt shipborne observations (b).

## 3 Method

### 3.1 The selection of optimal frequency channels

Although lower frequencies tend to better estimate snow depth, we used all frequencies to find the optimal frequency channels. All available combinations were compared to the OIB airborne snow depth measurements, and only VV combinations were used, since they had better performance than the HH combinations (Rostosky et al., 2018). To reduce the effect of uneven OIB measurement distributions within the passive microwave ratiometer grid cells caused by their resolution difference, on the same day, one passive microwave ratiometer grid cell (i.e., 25 km × 25 km) should contain at

least 2500 OIB measurement points. To reduce the influence of outliers, only OIB snow depth data between the 5th and 95th percentiles were used. In order to minimize the potential influence of sea ice concentration, only grid cells with ≥ 75% sea ice concentration were used. After these data filters, OIB data in November 2010, November 2016 and October-November 2018 were removed. Since the used OIB data were obtained in October or November each year, air temperatures could be higher than the melting point and cause surface melting. To reduce the influence of the snow layer's liquid water on

brightness temperatures observed by passive microwave radiometers, we excluded brightness temperatures that were assumed to be affected by liquid water based on the 2 m air temperature (T2m) data from ERA5 reanalysis data. If the T2m in a single grid cell during one day or during at least 5 of the 10 preceding days, is higher than 0 ℃, the brightness temperatures were removed (Rostosky et al., 2018). The GR was calculated as follows (take GR(37/7) as an example):

$$GR(37/7) = \frac{Tb_{37} - Tb_7 - k_1(1-C)}{Tb_{37} + Tb_7 - k_2(1-C)} \tag{1}$$


$$k_1 = Tb_{37,OW} - Tb_{7,OW} \tag{2}$$

$$k_2 = Tb_{37,OW} + Tb_{7,OW} \tag{3}$$

Where $C$ is the sea ice concentration, $k_1$ and $k_2$ are correction terms for the open water contribution when the sea ice concentration is below 100%. $Tb_{37,OW}$ and $Tb_{7,OW}$ are the brightness temperatures over open water at 37 and 7 GHz, the brightness temperatures over open water for the different frequencies can be referred to Ivanova et al. (2015).

Table 1 shows the correlation and root mean square deviation (RMSD) between OIB snow depth measurements and individual GRs (including both AMSR-E and AMSR-2 GRs). The combination of GR (37/19) and GR (19/10) was also applied here, since it was considered optimal for Arctic snow depth estimation (Markus et al., 2006). Different weightings (i.e., 3:2 and 2:3) had no obvious influence on the estimation result, and a weighting of 1:1 was used here.

Table 1. The relationships (including RMSD and correlation coefficient) between the OIB snow depth and different GRs.

| GR | RMSD (cm) | Correlation coefficient | Number of grid cells |
|---|---|---|---|
| GR(37/24) | 9.22 | -0.61 | 740 |

| | | |
|---|---|---|
| GR (37/19) | 9.11 | -0.62 |
| GR (37/11) | 8.95 | -0.64 |
| GR (37/7) | 8.92 | -0.64 |
| GR (24/19) | 9.21 | -0.61 |
| GR (24/11) | 9.03 | -0.63 |
| GR (24/7) | 9.14 | -0.62 |
| GR (19/11) | 9.15 | -0.62 |
| GR (19/7) | 9.46 | -0.58 |
| GR (11/7) | 10.62 | -0.41 |
| $\dfrac{GR\ (37/19)+GR\ (19/10)}{2}$ | 8.96 | -0.64 |

Except for GR (11/7), most GRs had good correlations with OIB snow depth, with correlation coefficients of >0.57 and RMSDs of <10 cm. GR (37/7), GR (37/11) and $\dfrac{GR\ (37/19)+GR\ (19/10)}{2}$ had better performances, and GR (37/7) performed optimally across all evaluation indices; thus, in the following section, we used GR (37/7) to construct a new snow depth estimation equation.

**3.2 The derivation of new snow depth estimation equation**

Fig. 2a shows the scatter plot between the OIB snow depth and GR (37/7), detailed temporal information for these data can be found in Sections 2.2 and 3.1, the spatial coverage is shown in Fig.1a. The corresponding regression equation can be derived as follows:

$$SD_{GR(37/7)}(cm) = 26.7 - 411 \cdot GR(37/7) \tag{4}$$

SSMIS frequencies were not as low as those of AMSR-E/2, meaning GR (37/7) could not be used with SSMIS data. Because of this, we used GR (37/19), which was the best combination among frequencies no less than 19 GHz, as shown in Table 1. The corresponding equation is listed as follows (Fig. 2b, the same data for the derivation of Eq. (4) were used):

$$SD_{GR(37/19)}(cm) = 23.5 - 601 \cdot GR(37/19) \tag{5}$$

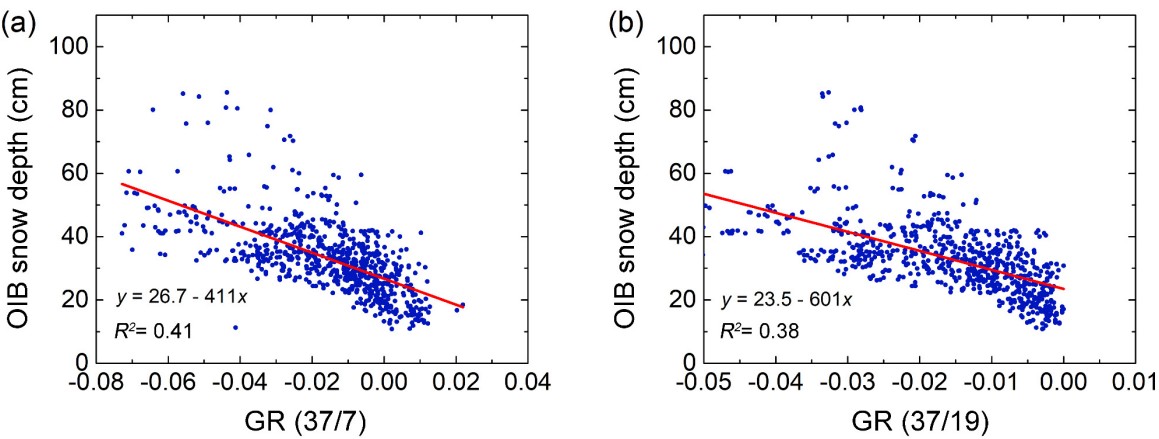

**Figure 2**. The scatter diagrams between the OIB snow depth and two GRs, i.e., GR (37/7) (a) and GR (37/19) (b).

To maintain the consistency of snow depth estimates based on two equations above, we compared their snow depth estimates during the OIB period (the same data for the derivations of Eqs. (4) and (5) were used). The snow depth estimations derived from the two equations agreed well and had an RMSD of 1.89 cm, and we corrected their original difference based on an empirical linear regression equation:

$$\text{SD}_{\text{GR}(37/7)}(\text{cm}) = \text{SD}_{\text{GR}(37/19)}(\text{cm}) - 0.03 \tag{6}$$

The snow depth from 1 October 2011 to 1 July 2012 was estimated based on Eq. (5) and Eq. (6), the snow depth for the remaining time periods was estimated from Eq. (4). Only valid snow depth estimates (> 0 cm) were allowed.

**3.3 The estimation of snow depth uncertainty**

The snow depth uncertainty was estimated from the uncertainty of individual input variables using Gaussian error propagation. Brightness temperature and sea ice concentration uncertainties were assumed to be 0.5 K and 5% (Rostosky et al., 2018). Uncertainties of the intercept and slope for Eqs. (4), (5) and (6) were 0.44 and 18.09, 0.57 and 27.95, 0.65 and 0.02, respectively. In addition, uncertainty due to the limited sample size of the OIB data should also be considered. A sensitivity analysis was performed to quantify the interannual variability (caused from the limited sample size) of the regression coefficients (see Section 4.1), the standard deviation of the regression coefficients deriving from different samples were assumed as the uncertainty value. These were ±3.23 cm for intercept and ±158.69 for slope. The uncertainties in the regression coefficients were summed to combine the uncertainties from linear fitting and limited OIB samples. Detailed calculation steps can be found in Rostosky et al. (2018).

Fig. 3 shows the spatial distributions of averaged snow depth uncertainty from 2002 to 2020 during four seasons: spring (October-December), summer (January-March), autumn (April-June) and winter (July-September) (Zwally et al., 2002). The snow depth uncertainty in summer (an average of 32.50 cm) was larger than that in the other seasons due to the effect of

liquid water in the snow layer. In autumn, winter and spring, the average snow depth uncertainties were approximately 20.76 cm, 17.85 cm and 23.79 cm, respectively. The averaged annual snow depth uncertainty was 23.73 cm. Spatially, smaller snow depth uncertainties were found in the sea ice interior, while larger uncertainties were found in the sea ice marginal region. As the sea ice concentration is the dominant factor affecting the observed brightness temperatures (and thus the GRs, Markus and Cavalieri, 1998), the influence of the open water is greater in the sea ice marginals and thus causes larger snow

depth uncertainties.

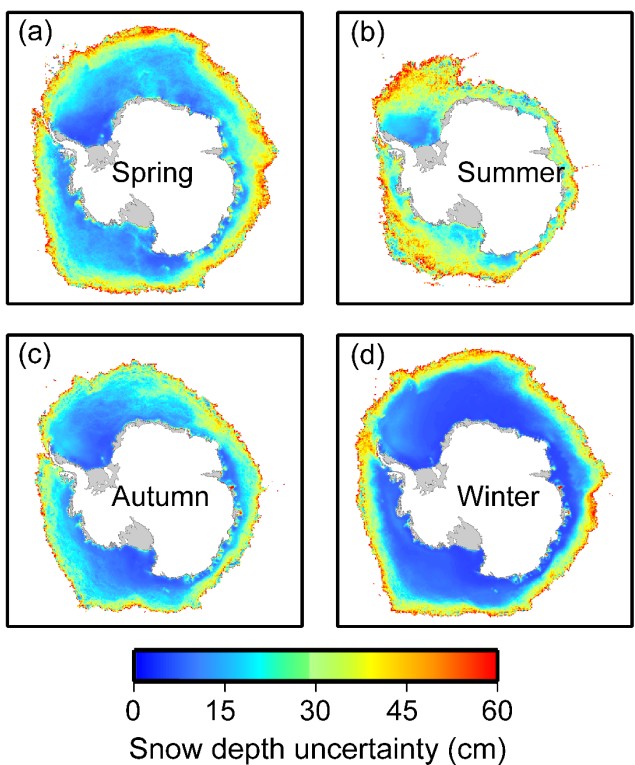

**Figure 3**. The spatial distributions of averaged snow depth uncertainty in different seasons from 2002 to 2020. Only grid cells with sea ice concentration ≥75% are shown here, grid resolution is 25 km.


## 4 Accuracy evaluation

### 4.1 Self-evaluation of the proposed method

To prove the robustness of the proposed method, we calculated 7 pairs of regression coefficients (for Eq. (4)) based on each 6-year combination of OIB snow depth data between 2009 and 2018 (Table 2). Only regression coefficients calculated from

more than 80 matched points were used for sensitivity analysis to ensure reliability. The uncertainty of the individual coefficient was estimated as its standard deviation. The estimated slope ranged from -474 to -349 with an uncertainty of

42.85, which caused a bias of <1 cm for the snow depth estimation; furthermore, the intercept varied from 25.4 to 28.3 with an uncertainty of 1.14. No obvious interannual variations could be found for either the slope or the intercept values.

Table 2. The regression coefficients of snow depth estimation equations based on OIB snow depth data in different years.

| Excluded year | Intercept | Slope | Number of grid cells |
|---|---|---|---|
| 2009 | 25.4 | -417 | 161 |
| 2010 | 27.2 | -445 | 88 |
| 2012 | 28.3 | -349 | 147 |
| 2013 | 23.8 | -707 | 40 |
| 2014 | 25.4 | -394 | 103 |
| 2016 | 27.3 | -474 | 134 |
| 2017 | 33.8 | -176 | 68 |
| All data | 26.7 | -411 | 740 |

Here, the OIB snow depth data in October 2016 were used to self-evaluate the snow depth estimation based on the equation derived from data in the remaining years (equation coefficients are shown in Table 2). The method proposed in Comiso et al. (2003) (hereafter called the Comiso method) was also applied for comparison, this is the commonly used snow

depth algorithm for Antarctic sea ice by using passive microwave radiometers, which modified the algorithm coefficients of the Markus and Cavalieri (1998) method to match the frequencies of AMSR-E. Data from October 2016 were chosen randomly, and the large size of this dataset ensured that the evaluation was comprehensive. In addition, the OIB data in individual years were independent, which also ensured the evaluation's objectivity. The result showed that the proposed method obviously outperformed the Comiso method with a mean difference of approximately -1.55 cm, while the latter had

an average difference of -19.15 cm, which greatly underestimated the snow depth (Table 3).

Table 3. The comparisons between the OIB snow depth and the snow depth estimates from our method and Comiso method in October 2016. MD: mean difference, MAD: mean absolute difference.

| | MD (cm) | MAD (cm) | RMSD (cm) | Correlation coefficient |
|---|---|---|---|---|
| Proposed method | -1.55 | 6.84 | 9.23 | 0.62 |
| Comiso method | -19.15 | 19.15 | 21.26 | 0.60 |

In addition, the snow depth derived from the proposed method had a narrower numerical distribution when compared to the OIB data (Fig. 4a). A peak of 30 cm could be found in both the proposed method and the OIB snow depth distributions; however, the Comiso method had a peak of 10 cm. This result confirms the conclusion of Worby et al. (2008a) that the Comiso method underestimates snow depth by a factor of 2.3. The snow depth estimated from the proposed method ranged from 20 to 60 cm, which was generally consistent with the OIB distribution. However, the OIB data had more snow depth

values of < 20 cm (Fig. 4a). A quadratic fitting equation was assumed to improve this situation; however, the uncertainties of the derived equation coefficients were usually larger (Rostosky et al., 2018).

Approximately 79% of the snow depth differences between the proposed method and OIB data had absolute differences of <10 cm, while the Comiso method showed that only 15% of the absolute differences were less than 10 cm, and 80% of the absolute differences were greater than 10 cm (Fig. 4b). Although the snow depths estimated from the proposed method and the Comiso method had almost the same variation pattern as the OIB snow depth data (here, Eq. (4) was used), the Comiso method obviously underestimated the snow depth by a mean difference of -17.3 cm at the interannual scale, nearly equal to the minimum OIB snow depth (Fig. 4c). Hence, compared to OIB snow depth measurements, the proposed method not only had a closer snow depth distribution but also showed a consistent temporal variation pattern, which demonstrated its reliability for estimating Antarctic snow depth.

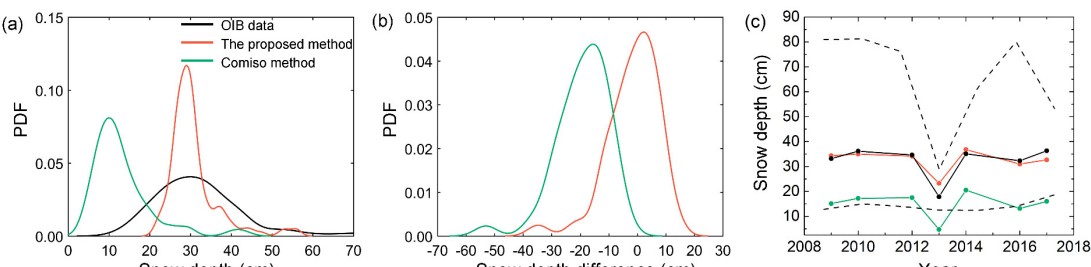

**Figure 4**. Comparisons of OIB snow depth to the snow depth estimates from both the proposed method and Comiso method. (a) The distributions of snow depth estimates in October 2016, (b) the probability density functions (PDFs) of differences between different snow depth estimates in October 2016 (red: the proposed method - OIB, green: Comiso method - OIB) and (c) the temporal variations of averaged snow depth estimates from 2009 to 2017 (red: the proposed method, green: Comiso method, black: OIB data). The black dashed lines in (c) show the variations in the maximum and minimum snow depth estimates from OIB.

Here the OIB snow depth data in October and November 2011 were also used to evaluate the snow depth estimates. On the one hand, OIB data in 2011 were not used for equation derivation (i.e., Eq. 4) and thus were suitable for an independent accuracy evaluation. On the other hand, in this case SSMIS-derived snow depth can be evaluated, which provides a reference for the performance evaluation of the Eqs. (5) and (6). The result showed that the proposed method still outperformed the Comiso method with a mean difference of -7.93 cm, while the latter still underestimated the snow depth with a mean difference of -24.65 cm (Table 4).

Table 4. The comparisons between the OIB snow depth and the snow depth estimates from the proposed method and Comiso method in October and November 2011. MD: mean difference, MAD: mean absolute difference.

| | MD (cm) | MAD (cm) | RMSD (cm) | Correlation coefficient |
|---|---|---|---|---|
| Proposed method | -7.93 | 10.63 | 13.81 | 0.27 |

| | | | | |
|---|---|---|---|---|
| Comiso method | -24.65 | 24.65 | 24.48 | 0.32 |

## 4.2 Comparison to AADC in situ measurements

As mentioned previously, liquid water in the snow layer affects observed brightness temperatures and causes larger uncertainties in estimated snow depths. The proposed method was thus mainly used for snow depth derivation in cold seasons, i.e., autumn, winter and spring. In this subsection, we focus on evaluating the performance of the proposed method in winter and spring. 1418 AADC snow depth measurement points were used for evaluation, and all AADC snow depth measurements within one passive microwave ratiometer grid cell were averaged and compared to passive microwave

ratiometer-derived snow depth (from the proposed method and Comiso method, respectively) in the same day. Each grid cell contained approximately 95 ± 36 AADC measurement points. The used AADC data were collected between September and October 2003 and between September and October 2007 (Fig. 1b).

The result showed that the proposed method performed better than the Comiso method across all evaluation indices (Table 5) with a mean difference of 5.64 cm, which was clearly less than the 14.47 cm value obtained using the Comiso method.

Although the number of AADC measurements were limited, the high accuracy and uneven distribution ensured the accuracy assessment was reliable and objective.

Table 5. The comparisons between the snow depth estimates from the proposed method and Comiso method and in situ measurements from AADC and ASPeCt. MD: mean difference, MAD: mean absolute difference. The results of the

evaluation by comparing to the ASPeCt data in the overlapped regions where both valid snow depth estimates from the proposed method and Comiso method can be found, are provided in brackets.

| | Comparison to AADC data | | Comparison to ASPeCt data | |
|---|---|---|---|---|
| | Proposed method | Comiso method | Proposed method | Comiso method |
| MD (cm) | 5.64 | -14.47 | 8.62 (8.94) | -9.96 (-10.16) |
| MAD (cm) | 10.77 | 17.08 | 13.80 (13.91) | 13.11 (13.20) |
| RMSD (cm) | 13.79 | 19.49 | 16.85 (16.85) | 17.61 (17.61) |
| Correlation coefficient | 0.42 | 0.40 | 0.13 (0.13) | 0.19 (0.19) |
| Number of grid cells | 15 | 15 | 264 (257) | 273 (257) |

## 4.3 Comparison to ASPeCt shipboard observations

Although the AADC in situ data were more accurate, their amount was still limited. To evaluate the methods at larger spatial

and temporal scales, ASPeCt shipboard observations were used for evaluation. All ASPeCt snow depth observations within one passive microwave ratiometer grid cell were averaged and compared to passive microwave ratiometer-derived snow depth (from the proposed method and Comiso method, respectively) in the same day. The operational periods of used

ASPeCt data are listed in Table 6, no data can be obtained in the missing periods. All ASPeCt data as listed in Table 6 were used here, and the spatial distribution is shown in Fig. 1b.


Table 6. The operational periods of used ASPeCt data in this study.

| Year | Month |
|------|-------|
| 2002 | August, September, December |
| 2003 | January, March, April, September, October |
| 2004 | March, April, October, November |
| 2005 | January, March, August, September |

Overall, the proposed method performed slightly better than the Comiso method, which was clearly different when compared to AADC data (Table 5). ASPeCt snow depth data are usually obtained from thin ice regions, as ships tend to
avoid thick ice; hence, the observed snow depth was with modal depths ranging between 0 and 10 cm (Worby et al., 2008b). In addition, the snow depth estimation equation from the Comiso method was derived from ASPeCt data, which might be unfair for the proposed method. Nevertheless, the proposed method still outperformed the Comiso method, and the similar performance demonstrated the reliability of the proposed method to estimate thin snow depth.

To show how the methods performed in different seasons, in Fig. 5 we compared the passive microwave ratiometer-
derived snow depth to the ASPeCt snow depth observations during four seasons. The performances of the two methods during the four seasons were different. The proposed method was most accurate during autumn with an RMSD of 0.10 m (13 grid cells, the same hereinafter), followed by summer with an RMSD of 0.15 m (73), winter with an RMSD of 0.18 m (65) and spring with an RMSD of 0.18 m (113), which were nearly comparable. For the Comiso method, accuracy was highest during winter with an RMSD of 0.10 m (65), followed by autumn with an RMSD of 0.14 m (11), summer with an RMSD of
0.15 m (77) and spring with an RMSD of 0.22 m (119). A larger bias was found in spring than in the other seasons. Overall, the proposed method was better than or comparable to the Comiso method during spring, summer and autumn.

A clear overestimation for the proposed method can be found by comparing to the ASPeCt data in all seasons (Fig. 5), we attribute this to the underestimation of ASPeCt snow depth observations. The thickness of level ice and snow cover, and estimates of surface ridging were recorded in the ASPeCt data, the latter was used to correct the level ice thickness based on
the mass of ice in ridges (Worby et al., 2008b). However, this correction was not be applied on snow depth observations, hence the ASPeCt snow depth data only represented for the level portions of ice floes (Worby et al., 2008a) and deformed ice with thicker snow cover was not included (Worby et al. ,2008b). As only thinner snow cover on level ice was included, it is obvious that these data underestimated the true snow depth (Worby et al., 2008a), which explains why the snow depth estimates from the proposed method were overall higher than these from ASPeCt data. In addition, the proposed method also
tends to overestimate the snow depth, as discussed in Sections 4.4 and 5.

Tables 7 and 8 show the seasonal evaluation of the mentioned two methods in Antarctic six sea sectors (Weddell West: 300°–315°, Weddell East: 315°–20°, Indian Ocean: 20°–90°, Pacific Sector: 90°–160°, Ross Sea: 160°–230°,

Bellingshausen-Amundsen Sea: 230°–300°). It should be noted that in some sea sectors, we could not construct the evaluations during all four seasons, due to the limited distribution of ASPeCt data. Both methods have lower accuracies in the Weddell West sector, which may be due to the thicker snow there, as the Weddell West is dominated by multiyear sea ice and it has similar emission signals to snow cover. Comparatively, the Comiso method underestimated snow depth in all sea sectors and seasons. Some negative correlation coefficients in Tables 7 and 8 can be found even in areas with comparably many grid cells, this is due to the observation bias of the ASPeCt data (± 20% bias was found for undeformed ice thicker than 0.3 m, and ± 30% bias for deformed ice, Worby et al. (2008b)). Due to the limited accuracy of ASPeCt samples, the evaluation may be biased, but ASPeCt shipborne data can still be assumed as a proxy for performance evaluation due to its large spatial-temporal coverage.

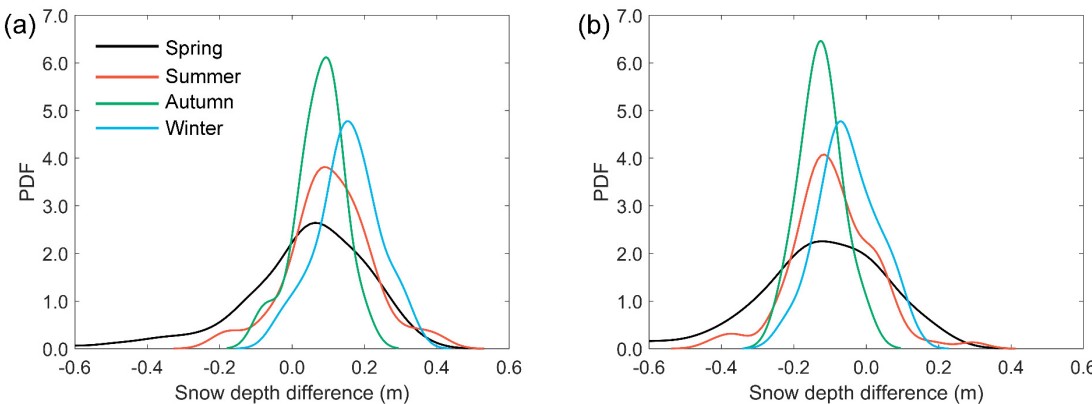

Figure. 5. The probability density functions (PDFs) of snow depth estimates differences of the proposed method (a) and Comiso method (b) by comparing to ASPeCt shipborne data in four seasons.

Table 7. The comparisons between the snow depth estimates from proposed method and in situ measurements from ASPeCt in different sea sectors and seasons. MD: mean difference, MAD: mean absolute difference.

|  | Season | MD (cm) | MAD (cm) | RMSD (cm) | Correlation coefficient | Number of grid cells |
|---|---|---|---|---|---|---|
| Weddell West | Spring | -0.07 | -0.13 | 0.19 | 0.30 | 21 |
|  | Summer | 0.15 | 0.19 | 0.23 | -0.03 | 10 |
| Weddell East | Spring | 0.13 | 0.17 | 0.19 | -0.37 | 48 |
|  | Summer | -0.10 | 0.10 | 0.10 | 1 | 1 |
|  | Winter | 0.14 | 0.14 | 0.14 | -0.39 | 16 |
| Indian Ocean | Spring | 0.12 | 0.12 | 0.12 | 0.49 | 3 |
| Pacific Sector | Spring | -0.05 | 0.15 | 0.20 | -0.16 | 26 |
|  | Autumn | -0.08 | 0.08 | 0.08 | 1 | 1 |
|  | Winter | 0.13 | 0.14 | 0.14 | 0.14 | 10 |
| Ross Sea | Spring | 0.02 | 0.05 | 0.06 | 0.31 | 15 |
|  | Summer | 0.10 | 0.12 | 0.14 | 0.07 | 62 |
|  | Autumn | 0.09 | 0.09 | 0.10 | 0.43 | 12 |

| | Season | | | | |
|---|---|---|---|---|---|
| Bellingshausen-Amundsen Sea | Winter | 0.17 | 0.17 | 0.19 | 0.20 | 39 |

Table 8. The comparisons between the snow depth estimates from Comiso method and in situ measurements from ASPeCt in different sea sectors and seasons. MD: mean difference, MAD: mean absolute difference.

| | Season | MD (cm) | MAD (cm) | RMSD (cm) | Correlation coefficient | Number of grid cells |
|---|---|---|---|---|---|---|
| Weddell West | Spring | -0.24 | 0.24 | 0.29 | 0.34 | 22 |
| | Summer | -0.06 | 0.15 | 0.18 | 0.45 | 11 |
| Weddell East | Spring | -0.02 | 0.11 | 0.15 | -0.31 | 49 |
| | Summer | -0.16 | 0.16 | 0.17 | 1 | 2 |
| | Winter | -0.08 | 0.08 | 0.09 | -0.50 | 16 |
| Indian Ocean | Spring | -0.03 | 0.05 | 0.07 | 0.85 | 4 |
| Pacific Sector | Spring | -0.27 | 0.27 | 0.33 | -0.16 | 24 |
| | Winter | -0.08 | 0.08 | 0.10 | 0.34 | 10 |
| Ross Sea | Spring | -0.12 | 0.13 | 0.15 | 0.28 | 20 |
| | Summer | -0.10 | 0.11 | 0.14 | 0.03 | 65 |
| | Autumn | -0.13 | 0.13 | 0.14 | 0.14 | 11 |
| Bellingshausen-Amundsen Sea | Winter | -0.03 | 0.08 | 0.10 | 0.24 | 39 |

**4.4 Comparison to satellite laser altimeter-derived snow depth data in both spatial and temporal scales**

Following Kern et al. (2016), we estimated the Antarctic snow depth in a complete year (January 2019 to December 2019)
from ICESat-2 using linear equations based on total freeboard, monthly snow depth estimates from ICESat-2 were then posted onto the 25 km grid cells and compared the results to estimates from the proposed method, as shown in Figs. 6 and 7. Generally, the spatial distribution patterns of snow depths estimated by the proposed method agreed closely with those derived from laser altimeters. Both snow depth datasets showed deeper snow cover mainly in the Weddell West and Bellingshausen-Amundsen Sea sectors.

As satellite laser altimetry is independent of the snow properties, satellite laser altimeters can better reveal snow depth evolution. Fig. 8 shows the monthly snow depth evolution in 2019 based on the two methods. Overall, the two snow depth time series were highly consistent and have an RMSD of 3 cm and a correlation coefficient of 0.86. Although the snow depth ranges from the two datasets still have some differences, the overall variation patterns were similar (except in summer). The existing differences in snow depth range and variation pattern were due to sensor and method differences. In clod seasons
(i.e., spring, autumn and winter), high consistency between the two datasets on both spatial and temporal scales implies the reliability of the proposed method.

An obvious snow depth overestimation for the proposed method can be found comparing to these from ICESat-2 in all months of 2019. Empirical linear regression models were used to compute snow depth from ICESat-2 total freeboard measurements. These empirical models were constructed based on the local sea ice measurements from 15 cruises in the

Southern Ocean over a time period of about 22 years (1986–2007). The limited coverage of this data set and the variable nature of snow cover over sea ice reduce the representativeness of this data set, which may contribute to the underestimation of snow depth estimates from ICESat-2, more local sea ice observation data (including snow depth, sea ice freeboard and sea ice thickness) are needed to improve the snow depth estimates from ICESat-2 in recent years (e.g., 2019).

    Snow depth retrieval based on passive microwave radiometers is sensitive to grain size (Markus and Cavalieri, 1998) and
ice type. For example, at microwave frequencies multiyear ice has a similar influence on the brightness temperatures as snow cover (Rostosky et al., 2018), and thus the snow depth over multiyear ice is overestimated in this case; in late winter/spring the grain size growth leads to a stronger reflected radiation and a reduction of the brightness temperature (Markus and Cavalieri, 1998). Both of these can influence the GRs and then increase the snow depth estimates. While satellite laser altimeters are independent of the snow properties and thus suffer less from the variable snow properties than passive
microwave radiometers. Thus, the difference between snow depth estimates from the passive microwave radiometers and ICESat-2 is due to their sensor and methodology difference, more observations of snow cover (including thickness, ice freeboard and snow properties) are needed to quantitatively explain the difference between these two snow depth estimates.

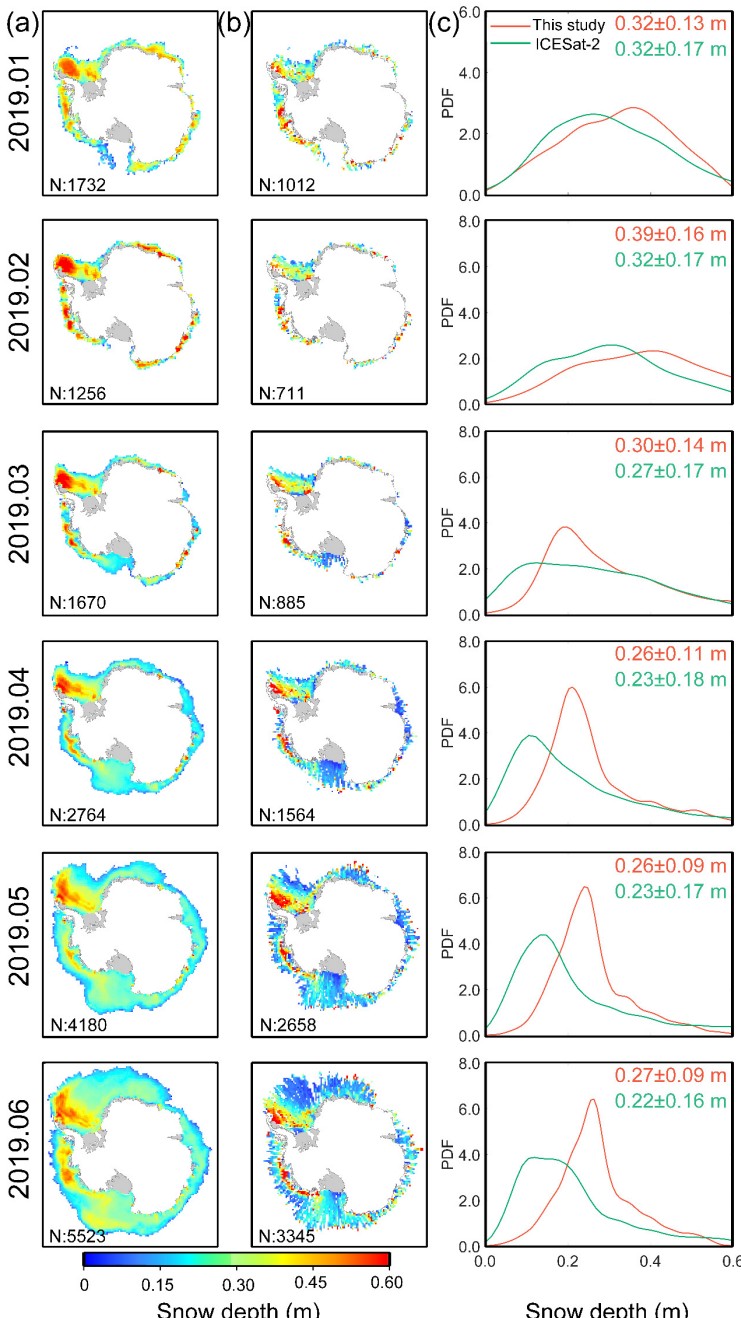

Figure 6. The spatial distributions of monthly snow depth estimates (January 2019 to June 2019) from this study (a) and ICESat-2 (b) together with the number (N) of valid grid cells in the bottom left corner of each image. (c) The probability density functions (PDFs) from (a) (red) and (b) (green). Numerical values in top right corner of (c) show the mean and standard deviation of the monthly snow depth estimates from (a) (red) and (b) (green).

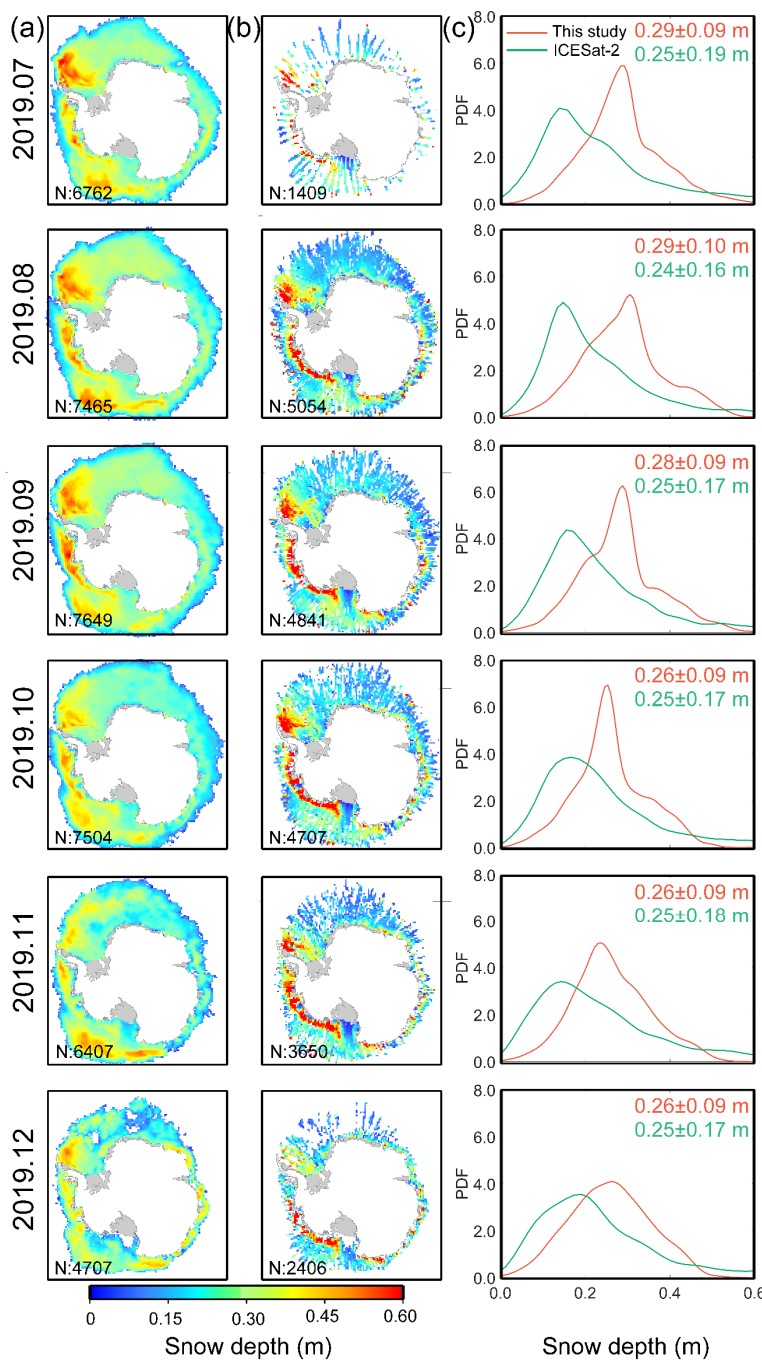


**Figure** 7. The spatial distributions of monthly snow depth estimates (July 2019 to December 2019) from this study (a) and ICESat-2 (b) together with the number (N) of valid grid cells in the bottom left corner of each image. (c) The probability

density functions (PDFs) from (a) (red) and (b) (green). Numerical values in top right corner of (c) show the mean and standard deviation of the monthly snow depth estimates from (a) (red) and (b) (green).

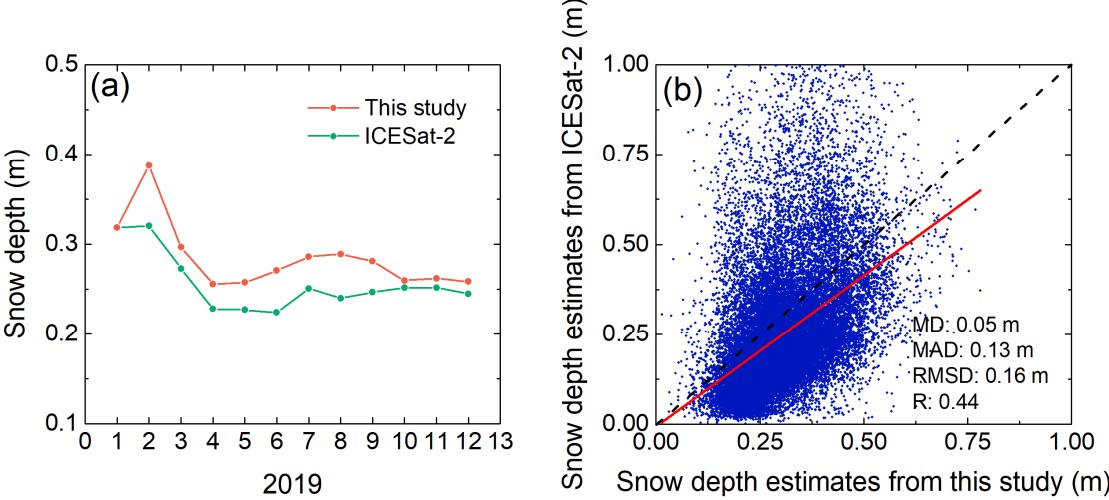

**Figure 8**. (a) Time series of the snow depth based on the proposed method in this study (red), and ICESat-2 (green) between January and December 2019. (b) Scatter diagrams of snow depth estimated from the proposed method and ICESat-2 in the grid scale. The line which fits to the scatter points is shown in red, the evaluation indices are also shown in the bottom right corner; MD: mean difference, MAD: mean absolute difference, R: correlation coefficient.

## 5 Spatio-temporal variation of Antarctic snow depth from 2002 to 2020

Although the proposed method was initially applied for snow depth estimation during the clod seasons (i.e., autumn, winter and spring), comparable performances were still found during summer; hence, we estimated the snow depth for all seasons from 2002 to 2020 and analysed the spatiotemporal variation pattern. The averaged Antarctic snow depth distributions from 2002 to 2020 showed obvious seasonal patterns (Fig. 9). In all four seasons, thin snow covers were seen in the marginal sea ice and thicker snow was located in the Weddell West and Bellingshausen/Amundsen Sea sectors, which was more obvious in summer. In winter, sea ice expands, and thicker snow cover could be found.

In Antarctic, cyclical thaw-freeze event can occur at all times (even in winter, Markus and Cavalieri, 1998); in summer due to this event the microwave signal changes rapidly within one day, and thus the brightness temperatures have large diurnal variations (Wankiewicz, 1993). This biases the snow depth estimates when daily averaged brightness temperatures are input to the snow algorithm. In summer the Antarctic is dominated by multiyear sea ice. Markus and Cavalieri (1998) found that in the Western Weddell Sea where perennial ice was present, daily variations of snow depth were higher in summer (January/February). As the thaw-freeze event can lead to larger grain sizes of snow cover, which can further result in an overestimation of snow depth, this event can cause large temporal variations in the snow depth estimates (Comiso et al., 2003). Due to the larger fluctuations of snow depth estimates in summer (Markus and Cavalieri, 1998), we infer that the

thaw-freeze event frequently occurs in summer and thus causes higher snow depth estimates in summer than in spring. This explains why snow depth increased from spring to summer during a melting period, and also explains why the thickest snow depth for the East Antarctic and Bellingshausen-Amundsen Sea is (where has a large amount of multiyear ice) in summer.

In addition, the emission signal of multiyear sea ice in microwave region is quite similar to these from snow (Rostosky et al., 2018), hence the snow depth on multiyear ice estimated from passive microwave radiometers is indeterminate and this

algorithm is more suitable for dry snow conditions (Comiso et al., 2003). Considering these, we suggest that it should be cautious when applying the proposed method for summer snow depth.

In particularly, we can find that the snow depth in the Weddell West sector decreased from autumn to winter during a growing period. As we mentioned before, the variability of grain size affects the brightness temperatures, in winter the grain size increases and thus results to a stronger radiation scattering. This effect can cause a reduced brightness temperature

(Markus and Cavalieri, 1998) and is stronger for higher frequencies (Rostosky et al., 2018), which leads to the overestimation of snow depth. The regression coefficients in proposed method were based on snow depth measurements in October and November (i.e., winter), considering the influence of the grain size on the microwave emission (Rostosky et al., 2018) these could lead to an overestimation of snow depth in autumn with smaller grain sizes. In the other sea sectors, this effect was not obvious, hence the proposed method still has the capability in estimating the snow depth distribution in

autumn. The above discussion indicates that in order to accurately retrieve snow depth over Antarctic sea ice in all seasons, more in situ observations of snow cover (including thickness and properties) with comprehensive spatio-temporal representativeness are needed to derive a more robust snow depth algorithm based on passive microwave radiometers, in the meantime the detailed understanding of the influences of snow properties (e.g. grain size and wetness) on brightness temperatures are also needed.

Antarctic snow depth showed a decreasing trend from 2002 to 2020 (Fig. 10a). This snow depth trend of Antarctic sea ice is the combined result from the six sea sectors, and the trend of snow depth may be "enhanced" or "offset". Hence, it is necessary to analyse the trend for individual sea sectors. All six sea sectors showed decreasing trends (Fig. 10), and these trends were decreasing across all four seasons.

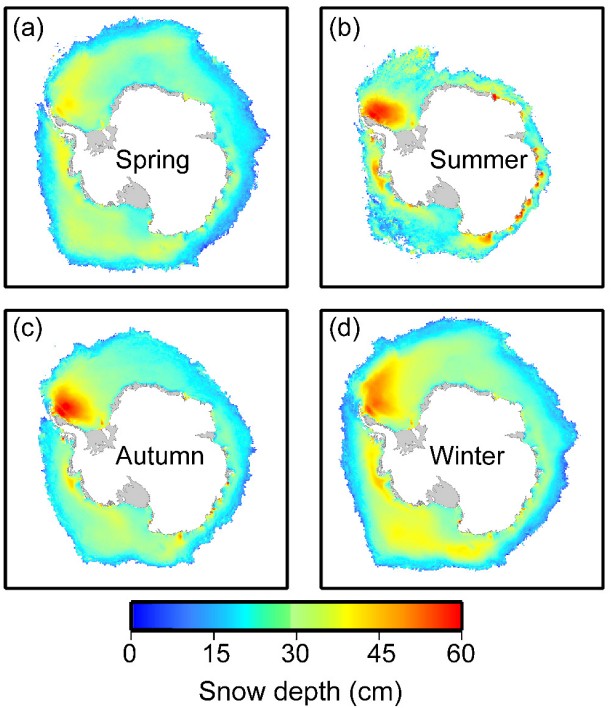


**Figure 9**. The spatial distributions of averaged snow depth in different seasons from 2002 to 2020. Only grid cells with sea ice concentration ≥75% are shown here, grid resolution is 25 km.

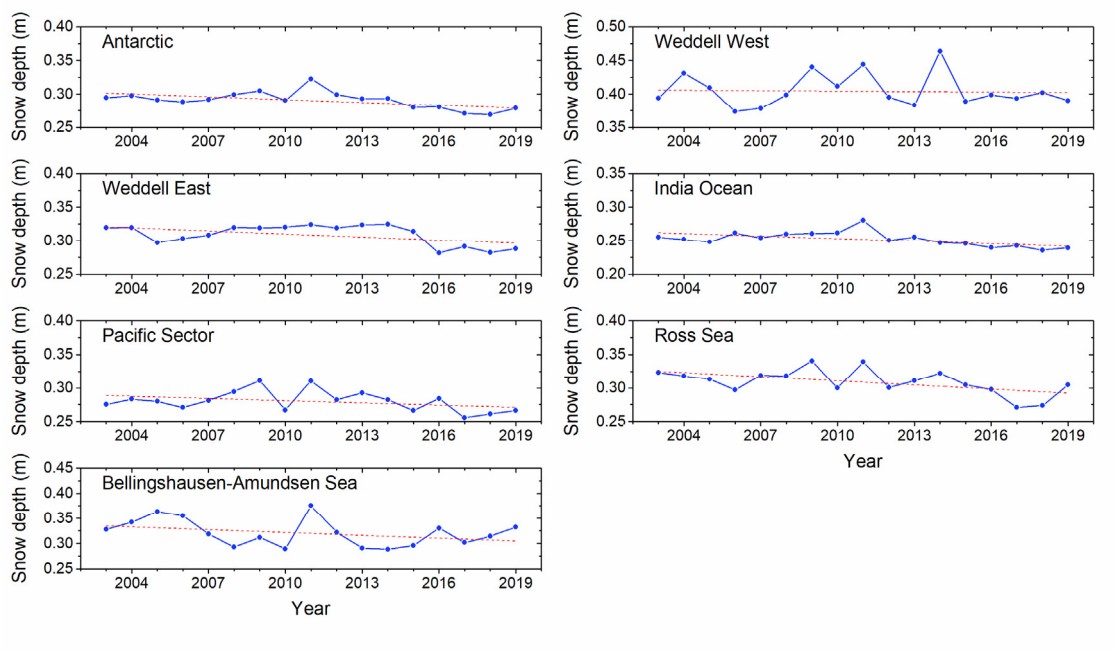

**Figure 10**. Time series of the annual snow depth estimates derived from the proposed method from 2003 to 2019 for the Antarctic and six sea sectors at the spatial resolution of 25 km.

The spatial distributions of snow depth variation trends during four seasons from 2002 to 2020 are shown in Fig. 11. During spring, except for the marginal sea ice in the western of Antarctic sea ice region, the snow depth in other regions showed clear decreasing trends. During summer and autumn, negative trends could be found in the Weddell West sector. During winter, a positive trend was found in the marginal sea ice of the Weddell West sector and West Ross Sea, while a decreasing trend was found in other sea sectors. In general, decreasing trends dominated the snow cover over Antarctic sea ice.

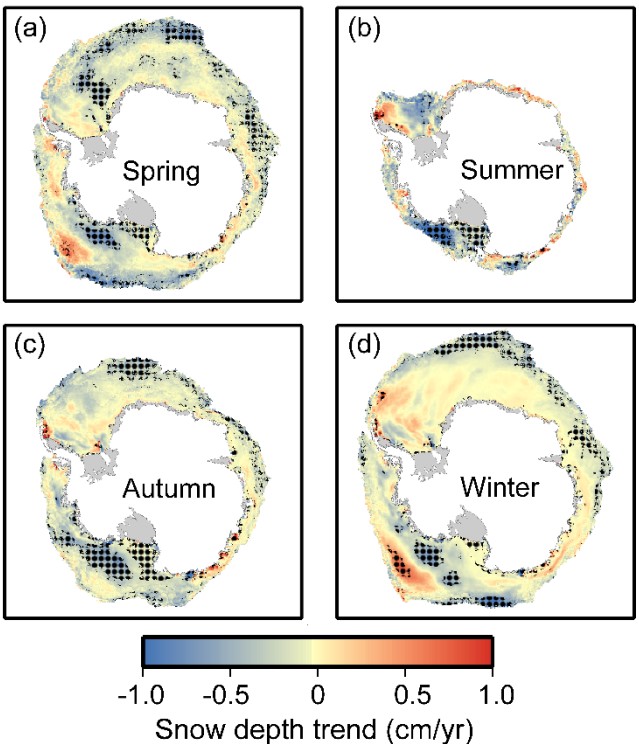

**Figure 11**. The spatial distributions of snow depth trends in different seasons from 2002 to 2020. For each grid cell, trend was only estimated when 12 years (or more) of snow depth estimates were obtained. Only grid cells with sea ice concentration ≥75% are shown here, grid resolution is 25 km. The black dot means that the trend is significant at 95% significance level according to two-tailed Student's t-tests.

## 6 Discussions

### 6.1 The uncertainty from estimation methods

Growth and melting of the snow layer will change the observed brightness temperatures; hence, the numerical relationship between brightness temperatures and snow depth is not fixed. When the snow layer starts to melt, its emissivity greatly differs from that of dry snow, which causes some biases in the snow depth estimation (Willmes et al., 2014). Hence, it is suggested that the proposed method in this study is limited to clod seasons (i.e., autumn, winter and spring). Although the used lower frequency suffers less from the volume scattering caused by the seasonal variation in the snow layer (such as densification or grain size increase, Rostosky et al., 2018), it is less sensitive to thin snow there may be some biases for snow depth in early winter. This result explains why the proposed method performed only slightly better than the Comiso method when compared with thinner ASPeCt snow depth. Since the snow depth over Antarctic sea ice is thicker than that in the Arctic and lower frequencies are more sensitive to thicker snow (Rostosky et al., 2018), this influence is assumed to be limited.

In addition, at the end of winter or early spring, the top snow layer melts during the day and refreezes at night. This forms an ice layer in the upper layer of snow cover. These ice layers usually have large grains which can contribute to the increase in scattering, and thus lead to overestimated snow depth (Markus and Cavalieri, 1998). In the Antarctic, the ice-covered snow layer may be covered by new snow (Willatt et al., 2009) and this melt–refreeze cycle will result in further overestimation. Besides, the relationship between brightness temperatures and snow depth is affected by snow density, snow grain size, flooded sea ice and weather conditions. However, because in situ measurements of these snow and ice properties are infrequently collected, their influences cannot be quantified and are thus not considered in the existing method. This issue can be solved with future in situ measurements. Lower frequencies are less affected by these factors and are more sensitive to deeper snow. Thus, they can improve the current Antarctic snow depth estimation. However, they are also more sensitive to roughness on the sea ice surface (Stroeve et al., 2006), and the spatial differences in snow emissivity derived from snow metamorphism in the Antarctic are rather small (Willmes et al., 2014). Nevertheless, similar performances of estimated snow depth with those derived from ICESat-2 at both spatial and temporal scales still demonstrate the reliability of the proposed method and imply that the snow depth uncertainties caused by the factors mentioned above are acceptable, as satellite laser altimeters are independent of snow properties.

Although the inclusion of low frequencies can reduce these influences, the linear regression equation may be too simple for some situations, e.g., very thin or thick snow. Some complex methods, e.g., polynomial fitting equations (Kilic et al., 2019), random forest regression models (Winstrup et al., 2019) and neural network models (Braakmann-Folgmann et al., 2019), may improve these situations. However, all these methods require more snow depth samples and much more training data if complex machine learning or deep learning technologies are used. Antarctic samples are quite sparse. Considering these potential limitations and the lack of a better operational snow depth product, we assume that the linear equation can estimate Antarctic snow depth more robustly in the current stage.

**6.2 The uncertainty from OIB data**

Since the derivation of the regression coefficients in Eq. (4) and Eq. (5) directly depends on the applied OIB samples, the uncertainty from the OIB data has a direct influence on snow depth estimation. Most of the OIB airborne measurements were taken in the western of Antarctic sea ice region during October or November, and their spatiotemporal representativeness was hence limited. Nevertheless, it was the most situatable data source for equation derivation considering its large spatial and temporal scales. The comparisons to ASPeCt and AADC data both demonstrated that this equation could be used in different seasons and sea sectors. Limited by the extreme climate and oceanic conditions in the Antarctic, in situ Antarctic sea ice measurement data are still limited. When more in situ data can be obtained, the corresponding algorithm can be further improved.

**6.3 The uncertainty from sea ice types**

It is still difficult for passive microwave radiometers to estimate snow depth over multiyear ice because multiyear ice scattering properties are similar to snow. On the one hand, reliable sea ice type data in the Antarctic was not accessible until now, and we still cannot derive snow depth estimation equations for different ice types. On the other hand, Antarctic sea ice is mostly young, first year ice, the amount of multiyear ice in the Antarctic is limited. This will be further improved when accurate Antarctic sea ice type data and in situ measurements are available.

**6.4 The uncertainty from applied spatial resolution**

Coarser spatial resolutions cannot obtain a detailed spatial pattern of snow depth. Although optical or SAR images have fine spatial resolutions, they still cannot estimate Antarctic snow depth on a daily scale. The passive microwave radiometer is one of the most effective sensors for daily Antarctic snow depth derivation. Although the current spatial resolution of the passive microwave radiometer is relatively coarse (25–50 km), considering the relatively flat surface of Antarctic sea ice and the urgent need for snow depth over Antarctic sea ice, the uncertainty caused by the coarser resolution is acceptable.

**6.5 The uncertainty from the different spatial resolutions of satellite, airborne, shipborne and field datasets**

Since the spatial resolutions of used data in this study are vastly different (i.e., satellite, airborne, shipborne and field data), the scale effect needs to be considered. Zhou et al. (2021) compared the snow depth values in various spatial grid cell spacings using OIB data, and found that the limited footprint of airborne data still caused the offset of snow depth values even when the coverage of these airborne measurements is overall good. Hence, the difference of spatial resolutions will further affect the comparison and evaluation of snow depth data. As the in situ measurements (including airborne, shipborne and field data) were not extensively obtained, i.e., the spatial coverage of these data in one satellite footprint is limited, investigation of how the spatial resolution of in situ data on snow depth comparation still could not be carried out at present. The reason is that airborne and shipborne data were usually obtained along the tracks, and the field measurements were

spatially sparse. This effect will be quantified in future work when much more in situ data are obtained. However, considering the sea ice cover in the Antarctic is relatively flat, the uncertainty caused by data sets with different spatial resolutions should be limited and smaller than that for Arctic sea ice.

## 7 Data availability

Snow depth product (including snow depth uncertainty) over Antarctic sea ice can be downloaded from National Tibetan Plateau Data Center, Institute of Tibetan Plateau Research, Chinese Academy of Sciences at http://data.tpdc.ac.cn/en/disallow/61ea8177-7177-4507-aeeb-0c7b653d6fc3/ (Shen and Ke, 2021, DOI: 10.11888/Snow.tpdc.271653). A short summary and some auxiliary information (including file naming, required software and etc.) are also provided.

## 8 Conclusions

Our study updates the regression equation for estimating snow depth over Antarctic sea ice using passive microwave ratiometers. By comparing 7-year OIB snow depth measurements, we found that the GR calculated from both lower and higher frequencies, i.e., GR(37/7), was best for deriving the Antarctic snow depth. It had an RMSD of 8.92 cm and a correlation coefficient of -0.64. The derived equation based on GR(37/7) was applied to consistent brightness temperatures from AMSR-E and AMSR-2. To fill the observation gaps between AMSR-E and AMSR-2, we used SSMIS data with a new equation based on GR(37/19) with a correction applied for consistent snow depth estimation. The estimated snow depth uncertainty analysis used a Gaussian error propagation. The mean uncertainty of the passive microwave ratiometer-derived snow depth was 23.73 cm.

The self-evaluation based on the combination of OIB data in different years showed that no obvious interannual variations could be found in the regression coefficients. The uncertainty of slopes from different combinations of OIB data was 42.85, which resulted in a snow depth estimation bias of <1 cm. The proposed method agreed well with the OIB data, showing a mean difference of -1.55 cm, and there was a similar snow depth variation pattern at the interannual scale. The Comiso method underestimated snow depth, with an average difference of -19.15 cm.

AADC data provided a comprehensive and unbiased assessment because they include measurements of both thick and thin snow layers. In comparison to AADC in situ measurements, the proposed method outperformed the Comiso method, with a smaller mean difference of 5.64 cm and an RMSD of 13.79 cm. The Comiso method underestimated snow depth with a mean difference of -14.47 cm and an RMSD of 19.49 cm.

The comparison to ASPeCt data showed that the proposed method had slightly better performance than the Comiso method (RMSDs of 16.85 cm and 17.61 cm, respectively) because the ASPeCt shipborne observations were focused on thin ice. The evaluation may be somewhat biased due to the observational accuracy of ASPeCt data (a mean bias of 20% or 30%).

Although the proposed method had better performance than the Comiso method, it could still be improved. We suggest that the proposed method should be used for the cold seasons, and it should be cautious when applying for summer snow

depth. Because a sufficient operational snow depth product is still lacking, we used our proposed method to generate a new, updated time series product of snow depth over Antarctic sea ice from 2002 to 2020 (including summer) on a daily scale. A decreasing trend of snow depth could be found in all six sectors and four seasons at the interannual scale. In addition, this dataset can be used to re-analyse data and acts as an input for sea ice thickness estimation.

## Author contribution

Xiaoyi Shen and Haili Li developed the related algorithm, generated and evaluated the snow depth product; Chang-Qing Ke supervised this work.

## Competing interests

The authors declare that they have no conflict of interest.

## Disclaimer

Publisher's note: Copernicus Publications remains neutral with regard to jurisdictional claims in published maps and institutional affiliations.

## Special issue statement

This article is part of the special issue "Extreme environment datasets for the three poles". It is not associated with a conference.

## Acknowledgments

AMSR-E brightness temperature data are derived at https://nsidc.org/data/AE_SI25/versions/3; AMSR-2 brightness temperature data can be derived at https://nsidc.org/data/AU_SI25/versions/1; SSMIS brightness temperature data are derived at https://nsidc.org/data/NSIDC-0001/versions/4; OIB airborne data can be derived at https://nsidc.org/data/ILATM2/versions/2; ASPeCt Data are derived at http://aspect.antarctica.gov.au/data; AADC in situ data can be derived at https://data.aad.gov.au/metadata/records/sea_ice_measurements_database. We thank all the data providers for their data.

## Financial support

This work is supported by the Programs for National Natural Science Foundation of China [grant numbers 41976212, 41830105].

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
