# Peer review of "Snow depth product over Antarctic sea ice from 2002 to 2020 using multisource passive microwave radiometers"

_Earth System Science Data, 2021_

## Author Comment (AC1)

We thank the reviewer for the helpful feedback, these suggestions have significantly improved the manuscript and figures, we are appreciative of his or her help and time.

We have addressed all the comments here, point by point responses to the comments are listed in BLUE.

General comments

The manuscript describes a new daily snow depth product on Antarctic sea ice derived from satellite passive microwave radiometry. The proposed methodology builds on previous algorithms, which were mainly applied to Arctic sea ice, and the authors first evaluate the most useful frequency combination for Antarctic sea ice and then derive the corresponding parameters. The method itself (using a gradient ratio of passive microwave measurements and fitting the linear coefficients) is rather trivial and has been employed previously (Markus and Cavalieri, Rostosky et al.), but previous products were not very accurate over Antarctic sea ice, which makes this manuscript and the corresponding dataset a novel and unique contribution. The authors conduct fairly extensive and comprehensive comparisons to a range of in-situ data as well as the Comiso method and ICESat-2 laser altimetry data to validate their method. The manuscript also analyses and visualises seasonally averaged mean snow cover and trends in the data.

This new dataset is a very welcome contribution to science on Antarctic sea ice and beyond, as currently no operational snow depth product is available for the Southern hemisphere. Therefore, I expect this dataset to be useful alone and in combination with e.g. satellite altimetry to derive sea ice thickness. The dataset covers the full lifetime of the AMSR-E and AMSR-2 sensors of 2002-2020 and employs SSMIS data to bridge the gap in between the two, making it a complete dataset. Updating the dataset regularly in the future would be very useful though.

The manuscript is well structured, clear and it has a good length. References and citations are mostly complete. A recent related study by Kacimi and Kwok (2020, https://doi.org/10.5194/tc-14-4453-2020) could be added, though. Figures and tables provide detailed insights into the dataset and the comparisons made for the accuracy assessment. Overall, they are presented in a comprehensive manner. A few details are too small and hard to see though. A time series of (e.g. average and per section) snow depth over the full 18 years would round the paper up and add to its credibility.

The method description is kept rather short, but clear enough, as references to previous studies are given to point the reader to more details. However, it might be worth writing down how the gradient ratio is calculated either in the appendix, on the website or in the readme file. What strikes me is that the derived uncertainties (which are provided with the dataset) are much smaller than the RMSDs calculated with respect to all other datasets. To me, the calculated RMSDs seem more realistic and I would therefore suggest rethinking and adjusting the uncertainty estimation, as the uncertainties are the values that are provided with the data.

In terms of the discussion, my view is that the derived trend and the comparison to the ASPeCt data might not be significant. On the other hand, I am missing a comment and discussion on the bias

between ICESat-2 and the proposed method. Please see more details in the specific comments.

The data is accessible via the given identifier and can be downloaded with an ftp client. Uncertainty estimates are included for each file. The data can be visualised and used for further analysis easily and the authors point to a range of open-source and standard software. The data description says 'all values are in meter', while snow depth is actually provided in 'cm' and so are the uncertainties. This might cause confusion, even if it should be clear from the data range.

Thank you very much for your feedback and advice, these suggestions have significantly improved the text and figures. We have revised the manuscript according to all your comments, details are listed in below.

Specific comments

Introduction: The study by Kacimi and Kwok (2020, https://doi.org/10.5194/tc-14-4453-2020) should be added, as it is quite related (same as general comment).

This reference has been added in the revised manuscript.

L. 42: You state that laser altimeters and a combination of laser and radar altimetry can be used to derive snow depth. However, also radar altimetry can be used stand-alone (without laser altimetry) to derive snow depth: See e.g. Guerreiro et al. (2016) 10.1016/j.rse.2016.07.013 and Lawrence et al. (2018) https://doi.org/10.5194/tc-12-3551-2018

Thank you for this important comment. These references (radar altimetry can be used to derive snow depth stand-alone) have been added in the text.

L. 57-58: Please rephrase the last part of the sentence. One could misunderstand it as all of the snow cover in Antarctica is thicker than 50 cm.

Agree. This sentence has been revised to avoid the existing misleading information:
'... this method is limited to dry snow less than 50 cm thick and thus may underestimate the snow depth in some regions of the Antarctic.'

The original sentence is:
'... , this method is limited to dry snow less than 50 cm thick, which is clearly less than the snow cover over Antarctic sea ice (Kwok et al., 2014).'.

L. 90: Delete 'than AMSR-E and AMSR2', otherwise the sentence makes no sense to me.

Accept and Done.

L. 93-94: How did you calibrate SSMIS? Please add more details.

The related reference has been added in the revised manuscript:

'*The SSMIS brightness temperature observations were calibrated to AMSR-E data based on the method from Wentz (2013), and ...*'.

Reference:

Wentz, F. J.: SSM/I Version-7 Calibration Report, Remote Sensing Systems, Santa Rosa, California, USA, RSS Technical Report 011012, 46 pp., 2013.

L. 95-96: Were (a) only SSMIS data used to calculate sea ice concentration in this time period, or (b) did you calculate sea ice concentration for the full time (2002-2020) yourself? Please clarify in the text.

If (a) is the case: Why did you choose the ARTIST algorithm rather than sticking to NASA Team, which is used for AMSR-E and AMSR2?

If (b) is the case: You don't need the downloaded sea ice concentration products mentioned before?! So, please modify the other text.

These brightness temperatures from AMSR-E, AMSR-2 and SSMIS were used to obtain the full time (2002-2020) sea ice concentrations by using the ARTIST Sea Ice (ASI) algorithm (i.e., (b)).

The related text has been revised as the sea ice concentration products do not need to be downloaded:
'*... Here, the AMSR-E/Aqua Daily L3 25 km Brightness Temperature  Polar Grids (Version 3) product ... were all applied. ... the NSIDC AMSR-E/AMSR-2 Unified L3 Daily 25 km Brightness Temperature  Polar Grids (Version 1) product was used. ...*'.

L. 109: 'subtracted' instead of 'misused'

Done.

L. 126: Please rephrase 'located in eastern and western Antarctica'.

The sentence has been revised:
'*... which were mainly **located in eastern and western of the Antarctic sea ice region (Fig. 1b).***'

L. 140: Why were these months excluded? Please clarify.

These months are excluded as no data are provided. This reason and the operation periods of used ASPeCt data have also been added in the revised text (Section 4.3 and Table 6):
'*... The operation periods of used ASPeCt data are listed in Table 6, no data can be obtained in the missing periods. ...*'.

Table 6. The operation periods of used ASPeCt data in this study.

| Year | Month |
|------|-------|
| 2002 | August, September, December |
| 2003 | January, March, April, September, October |
| 2004 | March, April, October, November |
| 2005 | January, March, August, September |

Fig. 1: In panel b the AADC 2007 data and 2005 ASPeCt data have almost the same colour (at least in my print out). Consider changing one of them.

The colour for AADC 2007 data has been changed in the Fig.1b. The updated figure is shown below:

[Figure]

Caption of Fig.1: I would mention AADC before ASPeCt to be consistent with the text and legend.

Accept and Done.

Eq. 1: Could be worth writing down how the gradient ratio is calculated either in the appendix or on the website / readme where the data can be found (same as general comment).

Also: Please specify (and cite) what you take as the brightness temperature over open water for the different frequencies.

The calculation equation for gradient ratio has been added into the revised text (in Section 3.1).

A reference has been added here to provide the brightness temperature over open water for the different frequencies:
'... *the brightness temperatures over open water for the different frequencies can be referred to Ivanova et al. (2015).*'.

L. 196-198 The calculated uncertainties are much smaller than the RMSDs calculated with respect to other datasets (same as general comment).

To give a more reliable uncertainty estimation, we have adjusted the uncertainty estimation method by adding the contribution of the limited sample size of the OIB data, this was constructed by performing a sensitivity analysis to quantify the interannual variability (caused from the limited sample size) of the regression coefficients from Eq. 4. The snow depth uncertainty in summer (an average of 32.50 cm) was larger than that in the other seasons due to the effect of liquid water in the snow layer. In autumn, winter and spring, the average snow depth uncertainties were approximately 20.76 cm, 17.85 cm and 23.79 cm, respectively. These values are closer to RMSDs calculated with respect to other datasets.

More details can be found in Section 3.3 in the revised text.

L. 282-284: It would be nice to indicate these sectors on the maps in Figure 1.

Agree and Done. The updated figure is shown below:

[Figure]

L. 283: Suggest renaming the 'Pacific Sector' or shifting the borders. 90 degrees is still in the middle of the Indian Ocean.

'Pacific Ocean' has been revised to 'Pacific Sector' in the revised text and figure.

Table 5: Overall, the correlation coefficients seem to be scattered around zero and even in areas with comparably many grid cells negative coefficients occur. This makes me doubt a physical correlation between the two datasets and question to which extend this comparison actually adds to the credibility of the proposed method. I therefore suggest either clarifying and discussing what could cause negative correlations or consider leaving this comparison out.

A discussion has been added here to clarify the negative correlations:
'*Some negative correlation coefficients in Tables 7 and 8 (original Tables 5 and 6) can be found even in areas with comparably many grid cells, this is due to the observation bias of the ASPeCt*

*data (± 20% bias is found for undeformed ice thicker than 0.3 m, and ± 30% bias for deformed ice, Worby et al. (2008b)). Due to the limited accuracy of ASPeCt samples, the evaluation may be biased, but ASPeCt shipborne data can still be assumed as a proxy for performance evaluation due to its large spatial-temporal coverage.'.*

Fig. 6+7: The numbers in panel c are extremely hard to read. Please increase the font size.

The font size of numbers in panel c of Figs. 6 and 7 has been increased.

In the caption of Fig. 6 and 7 panels a and b should be mentioned explicitly.

In the captions of Figs. 6 and 7, panels a and b have been mentioned explicitly in the revised text (take Fig. 6 as an example):
'*Figure 6. The spatial distributions of monthly snow depth estimates (January 2019 to June 2019) from this study (a) and ICESat-2 (b) together with the number (N) of valid grid cells in the bottom left corner of each image. (c) The probability density functions (PDFs) from (a) (red) and (b) (green). Numerical values in top right corner of (c) show the mean and standard deviation of the monthly snow depth estimates from (a) (red) and (b) (green).'.*

Figures 6+7c: The PMW approach seems to estimate higher snow depths for all months. Do you have an explanation for this bias? As this is quite striking, it should be mentioned in the text and ideally discussed what might cause it.

We have added a discussion to explain this overestimation when comparing with ICESat-2 as shown in Figures 6 and 7:

'*An obvious snow depth overestimation for the proposed method can be found comparing to these from ICESat-2 in all months of 2019. Regional empirical linear regression models were used to compute snow depth from ICESat-2 total freeboard measurements. The empirical linear regression models were constructed based on the local sea ice measurements from 15 cruises in the Southern Ocean over a time period of about 22 years (1986–2007). The limited coverage of this data set and the variable nature of snow cover over sea ice reduce the representativeness of this data set, which may contribute to the underestimation of snow depth estimates from ICESat-2, more local sea ice observation data (including snow depth, sea ice freeboard and sea ice thickness) are needed to improve the snow depth estimates from ICESat-2 in recent years (e.g., 2019).*

*Snow depth retrieval based on passive microwave radiometers is sensitive to grain size (Markus and Cavalieri, 1998) and ice type. For example, at microwave frequencies multiyear ice has a similar influence on the brightness temperatures as snow cover (Rostosky et al., 2018), and thus the snow depth over multiyear ice is overestimated in this case; in late winter/spring the grain size growth leads to a stronger reflected radiation and a reduction of the brightness temperature (Markus and Cavalieri, 1998). Both of these can influence the GRs and then increase the snow depth estimates. While satellite laser altimeters are independent of the snow properties and thus suffer less from the variable snow properties than passive microwave radiometers. Thus, the difference*

*between snow depth estimates from the passive microwave radiometers and ICESat-2 is due to their sensor and methodology difference, more observations of snow cover (including thickness, ice freeboard and snow properties) are needed to quantitatively explain the difference between these two snow depth estimates.'*

L. 337: What is the uncertainty of this trend? Upscaling 0.13cm/year is still only 2.3 cm over the full 18 years that you looked at, and smaller than the uncertainties that you found, so I wonder if it is actually a significant trend.

The uncertainty of this trend is 0.05 cm. A decreasing of 2.3 cm over the full 18 years was found and this value is smaller than the estimated snow depth uncertainty, hence the estimated trend may not be a significant trend. According to your comment below, we delete this table (i.e., Table 7) and show the time series of snow depth develops over the 18 years in the Antarctic and six sea sectors, which should be more useful.

Table 7: Instead or in addition to this table, I would find it really useful and interesting to see a time series of how snow depth develops over the full 18 years. You could for example plot the Antarctic wide mean and means from the sectors. Then you could also plot the calculated trend line on top. I think this would greatly increase the credibility of your dataset and the calculated trend, but could also verify the consistency of the three different satellites involved.

Time series of snow depth estimates over full 18 years in the Antarctic and six sea sectors has been added here and the original Table 7 has been removed. The same conclusion can be derived from this figure, i.e., Antarctic snow depth showed a decreasing trend from 2002 to 2020; all six sea sectors showed decreasing trends, and these trends were decreasing across all four seasons. The above statement has also been included in the revised manuscript. This figure is also listed below:

[Figure]

Figure 10. Time series of the annual snow depth estimates derived from the proposed method from

2003 to 2019 for the Antarctic and six sea sectors at the spatial resolution of 25 km.

Fig. 10: The black dots are almost impossible to see.

Also Fig. 10: I would suggest flipping the colour map to make it consistent with previous figures and common practices where blue corresponds to low numbers (decrease) and red to high numbers (increase), but this might be a personal preference.

Accept, the colour map has been flipped in the updated figure, and the size of black dots has been increased:

[Figure]

Figure 11. The spatial distributions of snow depth trends in different seasons from 2002 to 2020. For each grid cell, trend was only estimated when 12 years (or more) of snow depth estimates were obtained. Only grid cells with sea ice concentration ≥75% are shown here, grid resolution is 25 km. The black dot means that the trend is significant at 95% significance level according to two-tailed Student's t-tests.

L. 367: 'performed ONLY slightly better'

Done.

L. 467: Please check and update the link. "https://nsidc.org/data/G10011/versions/2" leads me to "On-Ice Arctic Sea Ice Thickness Measurements by Auger, Core, and Electromagnetic Induction, from the Late 1800s Onward, Version 2", which only contains data from the Arctic.

https://data.aad.gov.au/metadata/records/sea_ice_measurements_database

The link for AADC data has been corrected in the revised manuscript:
'*AADC in situ data can be derived at*
*https://data.aad.gov.au/metadata/records/sea_ice_measurements_database*'.

Data: The data description says 'all values are in meter', while snow depth is actually provided in 'cm' and so are the uncertainties. This might cause confusion, even if it should be clear from the data range (same as general comment).

We will revise the data description to give the right statement.

Technical corrections

L. 195: '-' missing between 'July-September'

Done.

L. 311: 'time series' instead of 'times'?!

Done.

L. 430: … showed THAT no obvious …

Done.

---

## Author Comment (AC2)

We thank the reviewer for the helpful feedback, these suggestions have significantly improved the manuscript and figures, we are appreciative of his or her help and time.

We have addressed all the comments here, point by point responses to the comments are listed in BLUE.

General Comments:

This paper compared Gradient Ratios and selected the best 37/7 combination of passive microwave radiometer AMSR-E/2 to derive the snow depth over Antarctic sea ice with a linear regression equation based on the Operation IceBridge (OIB) airborne snow depth measurements. Then compared the proposed method with the Cosmic method proposed in 2003 and validated the derived snow depth with self-evaluation of OIB, a few in-situ measurements AADC, shipborne measurements ASPeCt, and the laser altimetry ICESat-2 derived snow depth. After that a daily snow depth data of Antarctic sea ice from 2002 to 2020 (with gap filled with SSMIS) was produced. This work is of great value to derive snow depth over the Antarctic sea ice for nearly 20 years. However, it can be greatly improved after a major revision. My detailed comments are as follows:

Specific Comments:

This work lacks specific month information for all data used throughout the paper, such as in the data description and validation sections.

The specific month information for all data in the data description and validation sections have been added into the revised text (including literal statements and Tables).

About the growing and melting seasons, they should be clearly defined in the paper. I'm still doubt the suitability of the proposed method for both growing and melting seasons since the Eq.1 was derived from the growing season and the evaluation for supporting the suitability was with Aspect and ICESat-2 data. Furthermore, the model values have a clear overestimation than ICESat-2 in Fig.6 and 7.

A clear definition of the growing and melting seasons for Antarctic snow depth is not found from the currently published literatures, hence in the revised manuscript we delete the related statements and use the specific season (i.e., spring, summer, autumn and winter) instead.

Agree, we think that it is still doubtful to estimate the snow depth in summer using the proposed method, as shown in the unrealistic snow depth distribution in summer (Fig. 9b). We have added the related discussions (about the application of the proposed method in summer) in Sections 4.4 and 5 and made a clear statement that it should be cautious to apply the proposed method on summer snow depth.

A discussion about the overestimation of the proposed method than ICESat-2 has been added in Section 4.4:

*'An obvious snow depth overestimation for the proposed method can be found comparing to these from ICESat-2 in all months of 2019. Regional empirical linear regression models were used to compute snow depth from ICESat-2 total freeboard measurements. The empirical linear regression models were constructed based on the local sea ice measurements from 15 cruises in the Southern Ocean over a time period of about 22 years (1986–2007). The limited coverage of this data set and the variable nature of snow cover over sea ice reduce the representativeness of this data set, which may contribute to the underestimation of snow depth estimates from ICESat-2, more local sea ice observation data (including snow depth, sea ice freeboard and sea ice thickness) are needed to improve the snow depth estimates from ICESat-2 in recent years (e.g., 2019).*

*Snow depth retrieval based on passive microwave radiometers is sensitive to grain size (Markus and Cavalieri, 1998) and ice type. For example, at microwave frequencies multiyear ice has a similar influence on the brightness temperatures as snow cover (Rostosky et al., 2018), and thus the snow depth over multiyear ice is overestimated in this case; in late winter/spring the grain size growth leads to a stronger reflected radiation and a reduction of the brightness temperature (Markus and Cavalieri, 1998). Both of these can influence the GRs and then increase the snow depth estimates. While satellite laser altimeters are independent of the snow properties and thus suffer less from the variable snow properties than passive microwave radiometers. Thus, the difference between snow depth estimates from the passive microwave radiometers and ICESat-2 is due to their sensor and methodology difference, more observations of snow cover (including thickness, ice freeboard and snow properties) are needed to quantitatively explain the difference between these two snow depth estimates.'.*

The scale effect needs to be considered, since the scale of AMSR-E/2 is 25 km grid, ASPeCt is 1 km around the icebreaker, while the footprint of OIB is within 10 meters, and the AADC is even smaller. How would the different scales influence this study?

We have added a subsection in Section Discussions to discuss this issue:

*'Since the spatial resolutions of used data in this study are vastly different (i.e., satellite, airborne, shipborne and in situ data), the scale effect needs to be considered. Zhou et al. (2021) compared the snow depth values in various spatial grid cell spacings using OIB data, and found that the limited footprint of airborne data still caused the offset of snow depth values even when the coverage of these airborne measurements is overall good. Hence, the difference of spatial resolutions will further affect the comparison and evaluation of snow depth data. As the in situ measurements (including airborne, shipborne and field data) were not extensively obtained, i.e., the spatial coverage of these data in one satellite footprint is limited, investigation of how the spatial resolution of in situ data on snow depth comparation still could not be carried out at present. The reason is that airborne and shipborne data were usually obtained along the tracks, and the field measurements were spatially sparse. This effect will be quantified in future work when much more in situ data are obtained. However, considering the sea ice cover in the Antarctic is relatively flat, the uncertainty caused by datasets with different spatial resolutions should be limited and smaller than that for Arctic sea ice.'*

L74. This part lacks the introduction of ICESat-2

The introduction of ICESat-2 has been added here.

L87.The overlapping tracks and inter-comparison time period of SSMIS and AMSR-E/2 is not clearly introduced. The similar problem in Eq.2 and Figure 2b was also not clearly stated.

SSMIS and AMSR-E/2 can provide brightness temperature observations for the whole Antarctic in daily scale, hence they have the same observation region. The temporal coverages of SSMIS and AMSR-E/2 have both been added into the revised text, hence the overlapping time period can be easily derived. Consistency correction has been applied to the three passive microwave ratiometer datasets by using the published methods (Wentz, 2013; Du et al., 2014), and inter-comparison between SSMIS and AMSR-E/2 is not constructed in this study. These have been added into the revised manuscript.

A statement about the used data for Eq.2 and Figure 2b has been added into the revised manuscript.

L97. Since the sea ice is continuously drifting, which can be several kilometers per day, Section 2.2 should show the overlapping strategy of different datasets from spatial and temporal perspectives. For example, were the OIB snow depth measurements averaged in the overlapping AMSR-E/2 25 km grid in the same day? Generally, how much time difference between the two datasets in the same day?

The overlapping strategy of different datasets from spatial and temporal perspectives has been added into the Section 2.2:
'… *For comparison purposes, the OIB snow depth measurements were averaged in the overlapping passive microwave radiometer grid cells (at the spatial resolution of 25 km) in the same day. Although the sea ice is continuously drifting, the time differences between the OIB and passive microwave ratiometer data are always less than one day, which can cause the sea ice drift of several kilometers. Comparing to the coarse spatial resolution of passive microwave ratiometers (i.e., 25 km), this effect can be ignored. More details can be referred to Section 3.1, this processing method was also applied for AADC and ASPeCt data (in Sections 4.2 and 4.3).*'.

The brightness temperature data are averaged daily in the used passive microwave ratiometer datasets, and no detailed time information are provided. Hence the time difference between them and OIB data cannot be calculated. However, as we mentioned above (also in the revised text), the time difference between the OIB and passive microwave ratiometer data is less than one day, which can cause the sea ice drift of several kilometers. Comparing to the coarse spatial resolution of passive microwave ratiometers (i.e., 25 km), this effect can be ignored.

L117. Why was the OIB data in 2011 not used? Is it because there is no overlapping with AMSR-E/2 in the year? How about the validation of SSMIS-derived snow depth with OIB? I didn't see it in Section 4 accuracy evaluation.

In October and November 2011 (operation period of OIB mission in 2011) only brightness temperature observations from SSMIS can be obtained. Although inter-mission has been performed between SSMIS and AMSR-E/2, OIB data in 2011 were not used here to reduce the potential effect of the inter-mission calibration, and SSMIS only provide 5% brightness temperature observations for the full time (2002-2020).

In the revised manuscript, the OIB snow depth data in October and November 2011 were also used to evaluate the snow depth estimation. The result showed that the proposed method still outperformed the Comiso method with a mean difference of -7.93 cm, while the latter still underestimated the snow depth with a mean difference of -24.65 cm (Table 4). More details can be found in Section 4.1.

L148. Does Section 3.1 used both AMSR-E and AMSR-2 GR to correlate with OIB snow depth measurements? Please make it clear.

Yes, and this has been added in Section 3.1 in the revised text.

L192. How about the uncertainty of intercept a and slope b over FYI and MYI respectively here?

As we mentioned in the Section 6.3, reliable sea ice type data in the Antarctic was not accessible until now, and we still cannot derive snow depth estimation equations for different ice types. Hence, the uncertainty of intercept a and slope b over Antarctic sea ice (ice type is not classified) is provided here.

L199. About "due to complex surface conditions here", could you give some evidence to support this statement? Such as show the seasonal spatial distribution of GR.

Sea ice concentration is the dominant factor to affect the observed brightness temperatures of the ocean surface. As only the brightness temperatures from snow cover on sea ice are needed, the open water from the signal should be excluded. In the sea ice marginal regions, sea ice concentration is usually lower and the influence of open water on brightness temperatures are greater, this will further affect the calculated GRs. Thus, we think that the larger snow depth uncertainties in the sea ice marginal region is due to the larger amount of open water.

Here we give a clearer statement to explain why larger uncertainties are found in sea ice marginal region. No clear evidence shows that complex surface conditions are found in sea ice marginal regions and thus we delete the original statement:
'*As the sea ice concentration is the dominant factor affecting the observed brightness temperature (and the GRs, Markus and Cavalieri, 1998), the influence of the open water is greater in the sea ice marginal region and thus causes larger snow depth uncertainties.*'.

L207-208. Why is it 6-year combination instead of 7 (2009, 2010, 2012-2014, 2016-2018, as stated in L117)? As we can see the data 2018 missing in Table 2, is it because the data in 2018 less than 80 matched points? I'm confused with the number of grid cells in Table 2. Is it the "matched points"?

If it is, why were 2013 and 2017 included?

In order to reduce the effect of uneven OIB measurement distributions within the microwave ratiometer grid cells caused by their resolution difference, one microwave ratiometer grid cell (i.e., 25 km × 25 km) should contain at least 2500 OIB measurement points. To reduce the influence of outliers, only OIB snow depth data between the 5th and 95th percentiles were used. In order to minimize the potential influence of sea ice concentration, only grid cells with ≥ 75% sea ice concentration were used. After these data filters, OIB data in November 2010, November 2016 and October-November 2018 were removed. These sentences have also been added into the revised text to provide a clear statement.

L217. About "the Comiso method", please state the rationality to compare the proposed method with a paper published in 2003, nearly 20 years ago.

A statement has been added here:
'*The method proposed in Comiso et al. (2003) (hereafter called the Comiso method) was also applied for comparison, this is the commonly used snow depth algorithm for Antarctic sea ice by using passive microwave radiometers, which modified the algorithm coefficients of the Markus and Cavalieri (1998) method to match the frequencies of AMSR-E*'.

L243. About Figure 4, is this comparison of data in 2016? Please clearly state the time here. Is Figure 4c the mean snow depth in each year? Why does the year 2013 have an obvious valley? Based on the lower black dashed line in Figure 4c, I wonder why OIB cannot detect snow depth less than ~12 cm.

A specific time stamping (October 2016) has been added here.

Figure 4c shows the mean snow depth in each year. As we can found in Fig. 1a, OIB data in 2013 were obtained over sea ice in the Ross Sea, while in other years OIB data were obtained in Weddell Sea and Bellingshausen-Amundsen Sea. Hence the lower snow depth values in 2013 may due to the regional difference.

In order to minimize the potential influence of sea ice concentration, only grid cells with ≥75% sea ice concentration were used for equation derivation (Sections 3.1 and 3.2), the snow cover in high-concentration ice region is usually thicker. OIB flight tracks usually covered the sea ice near the coast, the ice thickness and snow depth are also usually thicker. Hence the snow depth detected by OIB is generally thicker than 12 cm.

L293. We may see a clear overestimation for the proposed method in Figure 5, please explain it. We can also see this overestimation when comparing with ICESat-2 from Figures 6 and 7.

We have added a discussion here to explain the overestimation for the proposed method in Figure 5:
'*A clear overestimation for the proposed method can be found by comparing to the ASPeCt data in*

*all seasons (Fig.5), we attribute this to the underestimation of ASPeCt snow depth observation. The thickness of level ice and snow cover, and estimates of surface ridging were recorded in the ASPeCt data, the latter were used to correct the level ice thickness based on the mass of ice in ridges (Worby et al., 2008b). However, this correction was not applied on snow depth observations, hence the ASPeCt snow depth data only represent for the level portions of ice floes (Worby et al., 2008a) and deformed ice with thicker snow cover was not included (Worby et al. ,2008b). As only thinner snow cover on level ice was included, it is obvious that these data underestimate the true snow depth (Worby et al., 2008a), which explains why the snow depth estimates using proposed method are overall higher than these from ASPeCt data.'*

We have added a discussion to explain this overestimation when comparing with ICESat-2 from Figures 6 and 7:

'*An obvious snow depth overestimation for the proposed method can be found comparing to these from ICESat-2 in all months of 2019. Regional empirical linear regression models were used to compute snow depth from ICESat-2 total freeboard measurements. The empirical linear regression models were constructed based on the local sea ice measurements from 15 cruises in the Southern Ocean over a time period of about 22 years (1986–2007). The limited coverage of this data set and the variable nature of snow cover over sea ice reduce the representativeness of this data set, which may contribute to the underestimation of snow depth estimates from ICESat-2, more local sea ice observation data (including snow depth, sea ice freeboard and sea ice thickness) are needed to improve the snow depth estimates from ICESat-2 in recent years (e.g., 2019).*

*Snow depth retrieval based on passive microwave radiometers is sensitive to grain size (Markus and Cavalieri, 1998) and ice type. For example, at microwave frequencies multiyear ice has a similar influence on the brightness temperatures as snow cover (Rostosky et al., 2018), and thus the snow depth over multiyear ice is overestimated in this case; in late winter/spring the grain size growth leads to a stronger reflected radiation and a reduction of the brightness temperature (Markus and Cavalieri, 1998). Both of these can influence the GRs and then increase the snow depth estimates. While satellite laser altimeters are independent of the snow properties and thus suffer less from the variable snow properties than passive microwave radiometers. Thus, the difference between snow depth estimates from the passive microwave radiometers and ICESat-2 is due to their sensor and methodology difference, more observations of snow cover (including thickness, ice freeboard and snow properties) are needed to quantitatively explain the difference between these two snow depth estimates.*'

For Figures 6 (L316) and 7 (L321), I suggest to add grid information for each satellite and model values for column c.

The grid information for each satellite has been added in the Figs. 6 and 7.
The model values (i.e., the snow depth estimates in this study) has been included for column c. In order to avoid the misleading information, we have changed the figure legend by replacing 'PMW' by 'This study' to give an explicit statement.

L326. Figure 8a, please add model values for comparison. Figure 8b, I suggest to use grid-wise comparison instead of monthly mean scatter in order to show more details.

The model values (i.e., the snow depth estimates in this study) has been included in Figure 8a. In order to avoid the misleading information, we have changed the figure legend by replacing 'PMW' by 'This study' to give an explicit statement.

A grid-wise comparison has been added in Figure 8b, and monthly mean scatter has been removed. The updated figure is shown below:

[Figure]

Figure 8. (a) Time series of the snow depth based on the proposed method in this study (red), and ICESat-2 (green) between January and December 2019. (b) Scatter diagrams of snow depth estimated from the proposed method and ICESat-2 in the grid scale. The line which fits to the scatter points is shown in red, the evaluation indices are also shown in the bottom right corner; MD: mean difference, MAD: mean absolute difference, R: correlation coefficient.

L334-336. There are several weird phenomena in Figure 9 as follows:

Snow depth increased from spring to summer during a melting period.
Snow depth in the Weddell West sector decreased from autumn to winter during a growing period.
The thickest snow depth for the East Antarctic and Bell-Amundsen sea is in summer, the hottest time for the Antarctic.
These issues should be discussed.

A discussion has been added in the revised text for these issues:

*'In Antarctic, cyclical thaw-freeze event can occur at all times (even in winter, Markus and Cavalieri, 1998); in summer due to this event the microwave signal changes rapidly within one day, and thus the brightness temperatures have large diurnal variations (Wankiewicz, 1993). This biases the snow depth estimates when daily averaged brightness temperatures are input to the snow algorithm. In summer the Antarctic is dominated by multiyear sea ice. Markus and Cavalieri (1998) found that in the Western Weddell Sea where perennial ice is present, daily variations of snow depth are higher*

*in summer (January/February). As the thaw-freeze event can lead to larger grain sizes of snow cover, which can further result in an overestimation of snow depth, this event can cause large temporal variations in the snow depth estimates (Comiso et al., 2003). Due to the larger fluctuations of snow depth estimates in summer (Markus and Cavalieri, 1998), we infer that the thaw-freeze event frequently occurs in summer and thus causes higher snow depth estimates in summer than in spring. This explains why snow depth increased from spring to summer during a melting period, and also explains why the thickest snow depth for the East Antarctic and Bellingshausen-Amundsen Sea is (where has a large amount of multiyear ice) in summer.*

*In addition, the emission signal of multiyear sea ice in microwave region is quite similar to these from snow (Rostosky et al., 2018), hence the snow depth on multiyear ice estimated from passive microwave radiometers is indeterminate and this algorithm is more suitable for dry snow conditions (Comiso et al., 2003). Considering these, we suggest that it should be cautious when applying the proposed method on summer snow depth.*

*In particularly, we can find that the snow depth in the Weddell West sector decreased from autumn to winter during a growing period. As we mentioned before, the variability of grain size affects the brightness temperatures, in winter the grain size increases and thus results to a stronger radiation scattering. This effect can cause a reduced brightness temperature (Markus and Cavalieri, 1998) and is stronger for higher frequencies (Rostosky et al., 2018), which leads to the overestimation of snow depth. The regression coefficients in proposed method were based on snow depth measurements in October and November (i.e., winter), considering the influence of the grain size on the microwave emission (Rostosky et al., 2018) these could lead to an overestimation of snow depth in autumn with smaller grain sizes. In other five sea sectors, this effect is not obvious, hence the proposed method still has the capability in estimating the snow depth distribution in autumn. The above discussion indicates that in order to accurately retrieve snow depth over Antarctic sea ice in all seasons, more in situ observations of snow cover (including thickness and properties) with comprehensive spatio-temporal representativeness are needed to derive a more robust snow depth algorithm based on passive microwave radiometers, in the meantime the detailed understanding of the influences of snow properties (e.g. grain size and wetness) on brightness temperatures are also needed.'*

Reference:
Wankiewicz, A., Multi-temporal microwave satellite observations of snowpacks, Ann. Glaciol., 17, 155-160, 1993.

In addition, how did you decide the spatial extent of sea ice in Figure 9?

Only grid cells with sea ice concentration ≥75% are shown here, grid resolution is 25 km. These have also been added into the figure caption.

L354. How did you decide the spatial extent of sea ice in Figure 10?

For each grid cell, trend was only estimated when 12 years (or more) of snow depth estimates were

obtained. Only grid cells with sea ice concentration ≥75% are shown here, grid resolution is 25 km. These have also been added into the figure caption.

L372. How does the scattering intensity lead to the overestimation? Is this a possible reason for the Figure 9b overestimation of snow depth?

At the end of winter or early spring, the top snow layer melts during the day and refreezes at night. This forms an ice layer in the upper layer of snow cover. These ice layers usually have large grains which can contribute to the increase in scattering, and thus and lead to overestimated snow depth (Markus and Cavalieri, 1998).

The above discussion has been added in the revised manuscript to give a clear statement.

This may be a reason for the overestimation of snow depth in Figure 9b. This has also been included in the discussion about the overestimation of summer snow depth in the Section 5 (i.e., Figure 9b).

Technical Corrections:

L11. I suggest to add "passive" to "microwave radiometers". Please check it throughout the paper.

Accept and Done.

L20. Please specify the "previous method"

A statement has been added here to specify the 'previous method':
'… the previous method (i.e., linear regression model based on GR(37/19); with RMSDs of 16.85 cm and 17.61 cm, respectively).'.

L34. The citation "Giles et al., 2018" work mainly uses ERS-2 to retrieve ice elevation instead of snow depth. Another paper of Giles may be more appropriate to be cited here:

Giles, K. A., et al (2007), Combined airborne laser and radar altimeter measurements over the Fram Strait in May 2002, Remote Sens. Environ., 111, 182–194.

Accept and Done.

L35-36. "long times" to "a long time period".

Accept and Done.

L44. It is better to use "in the Antarctic" than "in Antarctica" since we focus on sea ice. Please check the phrase throughout the paper.

Accept and Done.

L45. "uploading" to "upper"

Accept and Done.

L59. The citation "Kern et al., 2016" is a wrong one. The right one concentrating on the factor value is in "Kern, S., et al. (2011), An intercomparison between AMSR-E snow-depth and satellite C-and Ku-band radar backscatter data for Antarctic sea ice, Annals of glaciology, 52(57), 279-290"

Accept and Done.

L60. "successor" to "its successor"

Accept and Done.

L65. "reliable for estimating snow depth" to "suitable for retrieving Antarctic snow depth"

Accept and Done.

L79. "in the Arctic and in Antarctica" to "in Arctic and Antarctic." Please check similar phrases throughout the paper.

Accept and Done.

L82. About "pre-processing, bias correction and quality control were all applied", specifically how? Any introduction or citations here would be clearer.

The citations to these processing for three passive microwave radiometer products have been added in the related text.

L90 "(lower than or equal to 19 GHz)" to "than 19 GHz"

Accept and Done.

L93. "in" to "to"

Accept and Done.

L103-105. The reference "Schenk et al., 1999" for urban elevation DEM measurements should be cited to show the elevation accuracy, thus it should be placed after "1 m and 0.1 m"

Accept and Done.

L109. "misusing" to "subtracting"

Accept and Done.

L160-164. Please give the equation of GR here.

Accept and Done.

L163. What weightings? It is unclear.

We have revised this sentence to give a clear statement:
'*Different weightings (3:2 and 2:3) had no obvious influence on the estimation result, and a weighting of 1:1 was used here*'.

L168-169. Is "the root mean square residual" RMSD? If it is, I see it is different from that in Table 1. What is the "standard deviations of the derived regression coefficients"?

It should be the mean square of residual, not the root mean square of residual (i.e., RMSD). A linear regression analysis between OIB snow depth and GRs was performed here, and an equation can be obtained as listed below:
$SD = a + b \cdot GR$
Here, $a$ and $b$ are the derived regression coefficients, their standard deviations can also be used to evaluate the fitting performance.

RMSD and Correlation coefficient in Table 1 have provided enough information for the selection of optimal frequency channels, in order to avoid the redundant information, this sentence has been deleted in the revised text.

L178. "…GR (37/19), which ranked next to GR (37/7), as shown in Table 1" is an inaccurate statement. GR (37/19) was next to 37/11 instead of 37/7. You can say GR (37/19) was the best combination among frequencies no less than 19 GHz.

Accept and Done.

L185-186. The statement is too simple. Here lacks more spatio-temporal information for data used.

The spatio-temporal information for the used data has been added here.

L229. "a factor of 2", it is 2.3 exactly.

Agree and Done.

L250. Please show, in average, how many AADC points are used to compute a 25 km grid mean values.

The average and standard deviation values of the number of AADC points in one 25 km grid have been added here:

'… *Each grid cell contains approximately 95 ± 36 AADC measurement points.*'.

L259. Why the number of grid cells different between the two methods? Please clearly state if data used in this table all from the melting seasons.

In Table 5 (original Table 4), all ASPeCt data (as listed in the new Table 6) were used, these data are from individual months in four seasons, not just melting seasons. This statement has been added in the revised text. As a data filter was applied for snow depth estimates from proposed method and Comiso method, in some grid cells no valid snow depth estimate can be obtained for proposed method or Comiso method, hence the numbers of grid cells are different between the two methods. In the Table 5, the results of the evaluation using the ASPeCt data in the overlapped regions where both valid snow depth estimates from the proposed method and Comiso method can be found, are provided in brackets. The same conclusion can be derived based on these data.

Table 5. The comparisons between the snow depth estimates from the proposed method and Comiso method and in situ measurements from AADC and ASPeCt. MD: mean difference, MAD: mean absolute difference. The results of the evaluation using the ASPeCt data in the overlapped regions where both valid snow depth estimates from the proposed method and Comiso method can be found, are provided in brackets.

|  | Comparison to AADC data | | Comparison to ASPeCt data | |
| --- | --- | --- | --- | --- |
|  | Proposed method | Comiso method | Proposed method | Comiso method |
| MD (cm) | 5.64 | -14.47 | 8.62 (8.94) | -9.96 (-10.16) |
| MAD (cm) | 10.77 | 17.08 | 13.80 (13.91) | 13.11 (13.20) |
| RMSD (cm) | 13.79 | 19.49 | 16.85 (16.85) | 17.61 (17.61) |
| Correlation coefficient | 0.42 | 0.40 | 0.13 (0.13) | 0.19 (0.19) |
| Number of grid cells | 15 | 15 | 264 (257) | 273 (257) |

Table 6. The operation periods of used ASPeCt data in this study.

| Year | Month |
| --- | --- |
| 2002 | August, September, December |
| 2003 | January, March, April, September, October |
| 2004 | March, April, October, November |
| 2005 | January, March, August, September |

L279. "in other seasons" to "in the other seasons"

Done.

L286. About "most of the multiyear ice is in the Weddell West", "the Weddell West is dominated by multiyear sea ice" is more appropriate here.

Agree and Done.

L357. "method" to "methods"

Done.

L424. Delete "is"

Done.

L425. "at" to "to"

Done.

L446. Add "than the Comiso method" after "performance"

Done.

L476. The wrong doi should be https://doi.org/10.5194/tc-13-2421-2019

Corrected.